# Sampling from a $k$-DPP without looking at all items

**Daniele Calandriello**[*]
DeepMind Paris
dcalandriello@google.com

**Michał Dereziński**[*]
University of California, Berkeley
mderezin@berkeley.edu

**Michal Valko**
DeepMind Paris
valkom@deepmind.com

## Abstract

Determinantal point processes (DPPs) are a useful probabilistic model for selecting a small diverse subset out of a large collection of items, with applications in summarization, stochastic optimization, active learning and more. Given a kernel function and a subset size $k$, our goal is to sample $k$ out of $n$ items with probability proportional to the determinant of the kernel matrix induced by the subset (a.k.a. $k$-DPP). Existing $k$-DPP sampling algorithms require an expensive preprocessing step which involves multiple passes over all $n$ items, making it infeasible for large datasets. A naïve heuristic addressing this problem is to uniformly subsample a fraction of the data and perform $k$-DPP sampling only on those items, however this method offers no guarantee that the produced sample will even approximately resemble the target distribution over the original dataset. In this paper, we develop $\alpha$-DPP, an algorithm which adaptively builds a sufficiently large uniform sample of data that is then used to efficiently generate a smaller set of $k$ items, while ensuring that this set is drawn exactly from the target distribution defined on all $n$ items. We show empirically that our algorithm produces a $k$-DPP sample after observing only a small fraction of all elements, leading to several orders of magnitude faster performance compared to the state-of-the-art. Our implementation of $\alpha$-DPP is provided at https://github.com/guilgautier/DPPy/.

## 1 Introduction

Selecting $k$ diverse items out of a larger collection of $n$ items is a classical problem in computer science which naturally emerges in many tasks such as summarization (select $k$ phrases) and recommendation (select $k$ articles/ads to show to the user). An increasingly popular approach to model and quantify diversity in this subset selection problem is that of determinantal point processes (DPPs). Given a set $[n] \stackrel{def}{=} \{1, \ldots, n\}$ of $n$ items and a target size $k$, one can define a DPP of size $k$ (known as a $k$-DPP) through an $n \times n$ posivite semi-definite (PSD) similarity matrix $\mathbf{L}$ (also known as the kernel matrix). The matrix $\mathbf{L}$ encodes the similarities between items, and the user must choose it so that $[\mathbf{L}]_{ij}$ is larger the more the $i$-th and $j$-th items are similar. Given $k$ and $\mathbf{L}$, we define $S \sim k\text{-DPP}(\mathbf{L})$ as a distribution over all $\binom{n}{k}$ index subsets $S \subseteq [n]$ of size $k$, such that $\Pr(S) \propto \det(\mathbf{L}_S)$ is proportional to the determinant of the sub-matrix $\mathbf{L}_S$ induced by the subset. DPPs have found numerous applications in machine learning, not only for summarization [32, 23, 21, 7] and recommendation [19, 8], but also in experimental design [17, 34], stochastic optimization [39, 35, 15], Gaussian Process optimization [26], low-rank approximation [18, 24, 16], and more (recent surveys include [29, 4, 11]). Note that early work on DPPs focused on a *random-size* variant, which we denote $S \sim \text{DPP}(\mathbf{L})$, where the

---

[*]Equal contribution.

subset size is allowed to take any value between $0$ and $n$, and the role of parameter $k$ is replaced by the expected size $\mathbb{E}[|S|] = d_{\text{eff}}(\mathbf{L}) \stackrel{def}{=} \text{tr}\left(\mathbf{L}(\mathbf{L}+\mathbf{I})^{-1}\right)$. The quantity $d_{\text{eff}}(\mathbf{L})$ is known in randomized linear algebra [2, 11] and learning theory [6] as the effective dimension. While random-size DPPs exhibit deep connections to many scientific domains [25], the *fixed-size $k$-DPPs* are typically more practical from a machine learning stand-point [28].

**Sampling from a $k$-DPP.** The first $k$-DPP samplers scaled poorly, as they all relied on an eigende-composition [28] of $\mathbf{L}$ taking $\mathcal{O}(n^3)$ time. Replacing the eigendecomposition with a Cholesky factor-ization can increase numerical stability [30], and empirical performance [36] thanks to dynamically-scheduled, shared-memory parallelizations, but still ultimately require $\mathcal{O}(n^3)$ time. A number of methods have been proposed which use approximate eigendecomposition [31, 1] to reduce the com-putational cost, however these approaches provide limited guarantees on the accuracy of sampling.

To improve scalability, several approaches based on Monte-Carlo sampling were introduced, using rejection or Gibbs sampling. The fastest MCMC sampler for $k$-DPPs, to the best of our knowledge, is by [3] and has $n \cdot \text{poly}(k)$ complexity, i.e., asymptotically much faster than the cost of eigende-composition. However these MCMC methods do *not* sample exactly from the $k$-DPP distribution, and can only guarantee that the final sample will be close in distribution to a $k$-DPP. Moreover these guarantees only hold *after mixing*, which is difficult to verify and requires at least $\mathcal{O}(nk^2)$ time, making MCMC methods not applicable when $n$ is large.

A recent line of works [13, 10], using the ideas from [12, 14], developed sampling algorithms specially de-signed for a *random-size* DPP (as opposed to a $k$-DPP), which avoid expensive decomposition of the kernel while sampling exactly from $S \sim \text{DPP}(\mathbf{L})$. In particular, they showed that it is sufficient to first choose an intermediate subset $\sigma \subseteq [n]$ sampled i.i.d. from the *marginal* distribution of the DPP, i.e.,

| | Complexity |
|---|---|
| [28, 30, 36, 24] | $n^3$ |
| DPP-VFX [13] | $n \cdot k^{10} + k^{15}$ |
| $\alpha$-DPP (this paper) | $(\beta n \cdot k^6 + k^9)\sqrt{k}$ |

Table 1: Runtime comparison of exact $k$-DPP sampling algorithms. Here, $\beta \leq 1$ is the fraction of items observed by $\alpha$-DPP (see Theorem 1).

$\mathbb{P}(i \in \sigma) \approx \mathbb{P}(i \in S)$, and then sample from a DPP restricted to the items indexed by $\sigma$. Since the size of $\sigma$ is typically much less than $n$, this leads to a more efficient algorithm. Note that rescaling $\text{DPP}(\mathbf{L})$ into $\text{DPP}(\alpha\mathbf{L})$ using some constant $\alpha$ only changes the expected size of $S$ from $d_{\text{eff}}(\mathbf{L})$ to $d_{\text{eff}}(\alpha\mathbf{L})$. By accurately choosing an appropriate $\alpha_\star$, one can boost the probability that the random size of $S$ is exactly $k$, and convert a DPP sampler into a $k$-DPP sampler by repeatedly sampling $S \sim \text{DPP}(\alpha_\star\mathbf{L})$ until $S$ has size $k$. Based on this reduction, Dereziński et al. [13] gave the first algo-rithm (DPP-VFX) which is capable of exact sampling from a $k$-DPP in time $n \cdot \text{poly}(k)$. However, when sampling from $k$-DPPs, the approach of [13] has *two major limitations*:

1. DPP-VFX has an $\Omega(n)$ runtime bottleneck, since it requires computing all $n$ marginals, one for each item, in order to define the i.i.d. distribution of $\sigma$, which may be infeasible for very large $n$.

2. The reduction used by [13] to convert a DPP sampler into a $k$-DPP sampler increases the time complexity by a factor of at least $k^4$, resulting in a $\widetilde{\mathcal{O}}(n \cdot k^{10} + k^{15})$ runtime.

In this paper, we address both of these limitations by introducing a new algorithm called $\alpha$-DPP, which 1) does not need to compute all of the marginals, and 2) uses a new efficient reduction to convert from a random-size DPP to a fixed-size $k$-DPP (see Table 1 for comparison).

**Main contribution: uniform intermediate sampling for $k$-DPPs.** To resolve the $\Omega(n)$ runtime bottleneck, we use an additional intermediate sample $\rho$ based on *uniform* sub-sampling. Since uniform sampling can be implemented without looking at the actual items in the collection, this means that we do not even have to look at any item outside of $\rho$. The only necessary assumption required by our approach is that the maximum entry (i.e., similarity) of $\mathbf{L}$ is bounded by a constant $\kappa^2$. However, to simplify exposition we also assume w.l.o.g. that $d_{\text{eff}}(\mathbf{L}) \geq k$ (see Section 3).

In particular, we 1) sample $\rho$ uniformly out of $[n]$, then 2) only approximate the marginal probabilities of items in $\rho$ to compute $\sigma$, and finally 3) downsample $\sigma$ into a DPP sample $S$. To guarantee that $S$ is distributed exactly according to the DPP it is crucial that $\rho$ is diverse enough. We show that sampling a $k^2/d_{\text{eff}}(\mathbf{L})$ fraction of $[n]$ into $\rho$ (i.e., $|\rho| \approx k^2/d_{\text{eff}}(\mathbf{L}) \cdot n$) is enough. Since all the expensive computation is performed only on $\rho$, this gives us a $d_{\text{eff}}(\mathbf{L})/k^2$ speedup over existing methods.

**Theorem 1.** *Given any* $\mathbf{L} \succeq 0$ *with* $\max_{ij} \mathbf{L}_{ij} \leq \kappa^2$ *and* $1 \leq k \leq d_{\text{eff}}(\mathbf{L})$*, there exists an algorithm that returns* $S \sim k\text{-DPP}(\mathbf{L})$*, and with probability* $1 - \delta$ *runs in time*

$$\widetilde{\mathcal{O}}\big((\beta n \cdot k^6 + k^9)\sqrt{k}\log(1/\delta)\big),$$

*where* $\beta \leq \min\big\{k^2\kappa^2/d_{\text{eff}}(\mathbf{L}), 1\big\}$ *is the fraction of items observed by the algorithm.*

In the derivation of Theorem 1 we make several novel contributions. First, we provide a DPP sampler that given $\mathbf{L}$ and a rescaling $\alpha \leq 1$ leverages a mixture of uniform and rejection sampling to sample from $\text{DPP}(\alpha\mathbf{L})$ observing only an $\alpha\kappa^2 k$ fraction of the items. We then show that the optimal rescaling $\alpha_\star$ required by the reduction from $k$-DPP to DPP can be bounded with $\alpha_\star \leq \mathcal{O}(k/d_{\text{eff}}(\mathbf{L}))$, and thus our rescaling-aware sampler can sample from $k$-DPPs looking only at a $k^2/d_{\text{eff}}(\mathbf{L})$ fraction of the items. Finally, we provide an efficient search algorithm to find a close approximation $\hat{\alpha}$ of $\alpha_\star$.

**Model misspecification and computational free lunch.** Our result can be also interpreted from a perspective of model misspecification. Note that every time the users define a $k$-DPP they also implicitly define a random size $\text{DPP}(\mathbf{L})$. Moreover, the natural expected sample size (i.e., implicit number of unique items in $[n]$) of $\text{DPP}(\mathbf{L})$ is $d_{\text{eff}}(\mathbf{L})$, which does not depend on the desired size $k$. Therefore, if $\mathbf{L}$ is not chosen appropriately $d_{\text{eff}}(\mathbf{L})$ might be much larger than $k$, and the $k$-DPP is selecting $k$ unique items out of a much larger implicit pool of $d_{\text{eff}}(\mathbf{L}) \gg k$ unique items. In this case, it is possible to consider only a small $k^2/d_{\text{eff}}(\mathbf{L})$ fraction of the items selected uniformly at random and still have enough unique items to sample a diverse $k$-subset. Our result shows for the first time that it is possible to take advantage of this modeling disagreement between $k$ and $d_{\text{eff}}(\mathbf{L})$ to gain computational savings while still sampling *exactly* from the DPP, i.e., a computational free lunch.

**Binary search reduction from k-DPP to DPP.** Both our approach and the one of Derezinski et al. [13] rely on first implementing an efficient random-size DPP sampler, followed by the usage of a black-box construction based on rejection sampling to transform the DPP sampler into a $k$-DPP sampler. However the reduction of Derezinski et al. [13] requires access to a high-precision estimate of $d_{\text{eff}}(\alpha\mathbf{L})$ in order to appropriately tune $\alpha$. This makes optimizing $\alpha$ the bottleneck in the reduction from $k$-DPP to DPP, and therefore there is a large computational gap between the two problems. We close this gap thanks to a novel approach to find a suitable rescaling $\alpha$ based not on optimization but rather on binary search. Crucially, to find a suitable $\alpha$ this approach does not require an estimate of $d_{\text{eff}}(\alpha\mathbf{L})$, but only $\mathcal{O}(\sqrt{k}\log(n))$ black-box calls to a DPP sampler. Therefore, it can transform any random size DPP sampler into a $k$-DPP sampler with only a $\sqrt{k}$ overhead, and could be applied to any future improved sampler beyond this paper.

## 2 Sampling from a rescaled DPP with intermediate uniform subsampling

In this section we focus on a specific class of DPPs, $S \sim \text{DPP}(\alpha\mathbf{L})$, specified using a rescaling $\alpha \leq 1$ and a similarity matrix $\mathbf{L}$, which we refer to as rescaled DPPs. The main result of the section is showing that a sufficiently large subset selected uniformly at random can be used as an intermediate sample to sample from a rescaled DPP without looking at all of the items. The main reason to focus on rescaled DPPs is because they naturally appear when reducing $k$-DPP sampling to DPP sampling, where rescaling is used to align the random size of the DPP and $k$. This is going to be the focus of the next section. However the approach proposed in this section is not limited to rescaled DPPs, but under the right assumptions can be extended to accelerate sampling from generic DPPs. We will discuss these extensions at the end of the section.

**Notation** We use $[n]$ to denote the set $\{1, \ldots, n\}$. For a matrix $\mathbf{B} \in \mathbb{R}^{n \times m}$ and index sets $C, D$, we use $\mathbf{B}_{C,D}$ to denote the submatrix of $\mathbf{B}$ consisting of the intersection of rows indexed by $C$ with columns indexed by $D$. If $C = D$, we use a shorthand $\mathbf{B}_C$ and if $D = [m]$, we may write $\mathbf{B}_{C,[m]}$. Finally, we also allow $C, D$ to be multisets or sequences, in which case each row/column is duplicated in the matrix according to its multiplicity (and in the case of sequences, we order the rows/columns as they appear in the sequence). Note that with this notation if $\mathbf{L} = \mathbf{B}\mathbf{B}^\top$ then $\mathbf{L}_{C,D} = \mathbf{B}_{C,[n]}\mathbf{B}_{D,[n]}^\top$.

### 2.1 Background: distortion-free intermediate sampling.

Rather than sampling directly from the target DPP, intermediate sampling [10, 12] first selects an intermediate subset $\sigma$ from $[n]$, and then refines it by extracting $S$ from $\sigma$. Crucially, if $\sigma$ is selected according to a so-called Regularized DPP (R-DPP), this is equivalent to sampling $S$ from a DPP.

**Definition 1.** *For any psd matrix* $\mathbf{L} \in \mathbb{R}^{n \times n}$, *distribution* $p \overset{\text{def}}{=} \{p_i\}_{i=1}^n$ *and* $r > 0$, *define* $\widetilde{\mathbf{L}} \in \mathbb{R}^{n \times n}$ *with* $\widetilde{\mathbf{L}}_{i,j} \overset{\text{def}}{=} \frac{\mathbf{L}_{i,j}}{r\sqrt{p_i p_j}}$. *We define an* R-DPP$_p^r(\mathbf{L})$ *as distribution over events* $A \subseteq \bigcup_{k=0}^{\infty} [n]^k$ *such that*

$$\Pr(A) \overset{\text{def}}{=} \mathbb{E}_\sigma \left[ \mathbf{1}_{[\sigma \in A]} \det(\mathbf{I} + \widetilde{\mathbf{L}}_\sigma) \right] / \det(\mathbf{I} + \mathbf{L}), \quad for \quad \sigma = (\sigma_1, \ldots, \sigma_t) \overset{\text{i.i.d.}}{\sim} p, \quad t \sim \text{Poisson}(r).$$

**Proposition 1** ([10], Theorem 8). *For any* $\mathbf{L}$, $p$, $r$, *and* $\widetilde{\mathbf{L}}$ *defined as in Definition 1,*

$$if \quad \sigma \sim \text{R-DPP}_p^r(\mathbf{L}) \quad and \quad S \sim \text{DPP}(\widetilde{\mathbf{L}}_\sigma) \quad then \quad \{\sigma_i : i \in S\} \sim \text{DPP}(\mathbf{L}).$$

A computationally inefficient but conceptually simple approach to rejection sample $\sigma$ is the following:
1) compute all marginals $\mathbf{P}(i \in S) = \ell_i(\mathbf{L}) \overset{\text{def}}{=} [\mathbf{L}(\mathbf{I} + \mathbf{L})^{-1}]_i$ and sum to $\sum_{i=1}^n \ell_i(\mathbf{L}) = d_{\text{eff}}(\mathbf{L})$ [2];
2) sample $t \sim \text{Poisson}(c)$ and $\sigma \sim \text{Multinomial}\left(t, \frac{\ell_1(\mathbf{L})}{d_{\text{eff}}(\mathbf{L})}, \ldots, \frac{\ell_n(\mathbf{L})}{d_{\text{eff}}(\mathbf{L})}\right)$ for an appropriate constant $c$;
3) accept $\sigma$ w.p. $\frac{\det(\mathbf{I} + \widetilde{\mathbf{L}}_\sigma)}{C \det(\mathbf{I} + \mathbf{L})}$, where $C$ is an appropriate constant used to make the rejection step valid.

All existing intermediate sampling algorithms [10, 13, 12, 14] rely on this approach, refining it to make use of efficient approximations of the marginals $\ell_i(\mathbf{L})$ and adapting the constants $c$ and $C$ to the data. However they all share a common bottleneck: to sample $\sigma$ i.i.d. they need to approximate all marginals $\ell_i(\mathbf{L})$ and the normalization constant $d_{\text{eff}}(\mathbf{L})$, and therefore the final runtime scales as $n \cdot \text{poly}(k)$. While this is much smaller than the $\mathcal{O}(n^3)$ required by an exact sampler, it still becomes quickly unfeasible when $n$ is very large. In what follows we will introduce another approach to sample from an R-DPP that does not require to approximate the marginals of all items, but only the items selected in a preliminary *uniform* intermediate sample.

## 2.2 Faster DPP sampling with uniform intermediate sampling

We now introduce our novel $\alpha$-rescaled DPP sampler, called $\alpha$-DPP (see Algorithm 1). It requires as input a rescaling $\alpha$, a similarity matrix $\mathbf{L}$ and a parameter $r$ that will be used to tune the Poisson sampling step of Proposition 1 approach. It also requires as input a *dictionary* $\mathcal{D}$ containing $m$ elements, and set of weights stored in a diagonal matrix $\mathbf{W} \in \mathbb{R}^{m \times m}$. A dictionary is a subset of items $\mathcal{D} \subseteq [n]$ such that reweighting the items in $\mathcal{D}$ by $\mathbf{W}$ provides a good approximation of $\mathbf{L}$, so that the approximate marginals

$$l_i \overset{\text{def}}{=} \alpha[\mathbf{L} - \alpha\mathbf{L}_{[n],\mathcal{D}}^{\top}(\alpha\mathbf{L}_{\mathcal{D}} + \mathbf{W}^{-1})^{-1}\mathbf{L}_{[n],\mathcal{D}}]_i \quad (1)$$

computed using $\mathcal{D}$ and $\mathbf{W}$ are close to the true marginals $\ell_i$ (see Appendix E). Compared to the meta-approach of Proposition 1, the main technical difference is that rather than sampling directly $t$ from an appropriate Poisson, and then $\sigma$ from a Multinomial, we introduce an intermediate uniform sampling step. In particular, we first sample a Poisson $u$, and then uniformly sample a subset $\rho$ containing $u$ items. We

---

**Algorithm 1** $\alpha$-DPP sampler

**Input:** $\alpha$, $\mathbf{L}$, $\mathcal{D}$, $\mathbf{W}$, $r \geq 1$
1: Set $\widehat{\mathbf{L}} = \mathbf{W}^{1/2}\mathbf{L}_{\mathcal{D},\mathcal{D}}\mathbf{W}^{1/2} \in \mathbb{R}^{m \times m}$
2: **repeat**
3: $\quad$ Sample $u \sim \text{Poisson}(re^{1/r}\alpha n\kappa^2)$
4: $\quad$ Sample $\rho = \text{Uniform}(u, [n])$
5: $\quad$ **for** $j = \{1, \ldots, u\}$ **do**
6: $\quad\quad$ Compute $l_{\rho_j}$ using Eq. (1)
7: $\quad\quad$ Sample $z_j \sim \text{Bernoulli}(l_{\rho_j}/(\alpha\kappa^2))$
8: $\quad$ **end for**
9: $\quad$ Set $\sigma = \{\rho_j : z_j = 1\}$, $t = |\sigma|$
10: $\quad$ Set $[\widetilde{\mathbf{L}}_\sigma]_{ij} = \frac{1}{r\sqrt{l_{\sigma_i} l_{\sigma_j}}}[\mathbf{L}]_{\sigma_i\sigma_j}$
11: $\quad$ $Acc \sim \text{Bernoulli}\left(\frac{e^{d_{\text{eff}}(\alpha\widehat{\mathbf{L}})}\det(\mathbf{I} + \alpha\widetilde{\mathbf{L}}_\sigma)}{e^{t/r}\det(\mathbf{I} + \alpha\widehat{\mathbf{L}})}\right)$
12: **until** $Acc$ = true
13: Sample $\widetilde{S} \sim \text{DPP}(\alpha\widetilde{\mathbf{L}}_\sigma)$
14: **return** $S = \{\sigma_i : i \in \widetilde{S}\}$

---

then compute an approximation $l_i$ of the marginal $\ell_i$ only for the items in $\rho$, and downsample $\rho$ into $\sigma$ using rejection sampling (Line 7). Finally, we accept or reject $\sigma$ (Line 11) and then downsample $\sigma$ into $S$ using a standard DPP sampler on the smaller $\widetilde{\mathbf{L}}_\sigma$.

Algorithm 1 is not simply a different implementation of the approach of Proposition 1, since even if Multinomial sampling is implemented with lazy evaluations of $l_i$, we would still need to compute the *normalization* constant of the Multinomial, which strictly requires computing all $l_i$. Similarly, the rejection test of Line 11 is also designed to accept as many candidates as possible without requiring the computation of the normalization constant as in [13]. Rather our approach is a novel method to sample from an R-DPP using Poisson rejection sampling. In particular, we prove not only that it always returns an $S$ sampled according to the exact DPP distribution, but also that if the dictionary satisfies certain conditions, the main of which is $(\varepsilon, \alpha)$-accuracy (see Appendix E and [5]), then the algorithm will generate $S$ quickly.

**Theorem 2.** *Given any* $\mathbf{L} \succeq \mathbf{0}$, *dictionary* $\mathcal{D}$, $\mathbf{W} \succ \mathbf{0}$, $r \geq 1$ *and* $\alpha > 0$, $\alpha$-*DPP returns* $S \sim \mathrm{DPP}(\alpha\mathbf{L})$. *Moreover, if* $r \geq d_{\mathrm{eff}}(\alpha\mathbf{L}) \geq 1/2$, $\mathcal{D}$ *and* $\mathbf{W}$ *are* $(1/d_{\mathrm{eff}}(\alpha\mathbf{L}), \alpha)$-*accurate,* $\mathcal{D}$ *satisfies* $|\mathcal{D}| \leq 10 d_{\mathrm{eff}}(\alpha\mathbf{L})$, *and* $d_{\mathrm{eff}}(\alpha\widehat{\mathbf{L}}) \leq 10 d_{\mathrm{eff}}(\alpha\mathbf{L})$, *w.p.* $1 - \delta$ $\alpha$-*DPP runs in time*

$$\mathcal{O}\Big( \big[\min\{\alpha\kappa^2 d_{\mathrm{eff}}(\alpha\mathbf{L}), 1\} \cdot n \cdot d_{\mathrm{eff}}(\alpha\mathbf{L})^6 \log^2(n/\delta) + d_{\mathrm{eff}}(\alpha\mathbf{L})^9 \log^3(n/\delta)\big] \cdot \log(1/\delta)\Big).$$

The main implication of our result is that the intermediate distribution based on marginals can be replaced more and more accurately with a uniform distribution as $\alpha$ becomes smaller. This results in having to compute marginals only for a $\min\{\alpha\kappa^2 d_{\mathrm{eff}}(\alpha\mathbf{L}), 1\}$ fraction of the $n$ items. This speedup can be significant when the rescaling $\alpha$ is very small, as is the case when we want to sample a small number of items out of a large collection. Compared to other exact DPP samplers, such as DPP-VFX, our $\alpha$-DPP is strictly faster by roughly a $1/(\alpha\kappa^2 d_{\mathrm{eff}}(\alpha\mathbf{L}))$ factor when implemented with an appropriate caching strategy for the estimates $l_i$ (see Appendix E). Further, unlike MCMC samplers, $\alpha$-DPP is an *exact* sampler. Moreover, there is no known MCMC approach that can achieve a runtime sub-linear in $n$ when $\alpha$ is small as $\alpha$-DPP.

An $(\varepsilon, \alpha)$-accurate dictionary that also satisfies the other conditions can be generated using a slight modification of the BLESS algorithm [37], that we call BLESS-I algorithm, presented in Appendix C. However, note that since the marginals $\ell_i$ are equivalent to the ridge leverage scores [2] of item $i$, we can replace BLESS-I with any present or future algorithm for leverage score sampling that can be modified to be rescaling-aware [5, 37]. Moreover, note that BLESS-I also returns an estimate of $d_{\mathrm{eff}}(\alpha\mathbf{L})$ that is sufficiently accurate to tune $r$ and $\varepsilon$. At the same time, our analysis could be excessively conservative, and instead of trying to set $r$ and $\varepsilon$ using $d_{\mathrm{eff}}(\alpha\mathbf{L})$ as suggested by Theorem 2, a more practical strategy is to start with a constant $r$ and increase it slowly if the sampler is rejecting with a too low probability, using a doubling schedule to preserve overall time complexity.

**Proof sketch.** The proof is divided in two parts, proving that $\alpha$-DPP is an exact sampler (Lemma 6) and that under the right conditions it is efficient (Lemma 7).

For the first part we once again rely on the approach of Proposition 1, but with the added difficulty of not being allowed to compute all the marginals. To avoid this bottleneck, we show that:

    A) sampling $t \sim \mathrm{Poisson}(r)$ and $\sigma \sim \mathrm{Multinomial}(t, \{\ell_i/d_{\mathrm{eff}}(\alpha\mathbf{L})\}_{i=1}^n)$;    and

    B) sampling $n$ independent $s_i \sim \mathrm{Poisson}(r'\ell_i(\alpha\mathbf{L}))$, and adding $s_i$ copies of item $i$ to $\sigma$,

are equivalent for an appropriate choice of $r$ and $r'$, i.e., we prove that the $\sigma$ generated by both approach A and B follow the same distribution. However, unlike approach A, approach B does not require computing a normalization constant, i.e., it samples from *unnormalized* probabilities. Moreover, if we know an upper bound on the marginals we can further reduce the number of marginals that need to be computed. In our case we use the bound $\ell_i \leq \alpha\kappa^2$, and show that

    C) sampling $n$ Poisson independently $u_i \sim \mathrm{Poisson}(r'\alpha\kappa^2)$, only if $u_i > 0$ computing $\ell_i$ and sampling $s_i \sim \mathrm{Binomial}(u_i, \ell_i/(\alpha\kappa^2))$, and adding $s_i$ copies of item $i$ to $\sigma$

once again generates $\sigma$ strictly equivalent to the ones of approach B and A. The added advantage of approach C over the others is that only the marginals of items with $u_i > 0$ are actually computed, and there is no need to compute a normalization constant. Starting from this new approach, to obtain our $\alpha$-DPP sampler (Algorithm 1) we simply replace the $n$ Poisson $u_i \sim \mathrm{Poisson}(r'\alpha\kappa^2)$ with a single $u \sim \mathrm{Poisson}(r'\alpha\kappa^2 n)$ followed by uniform sampling, and replace the exact $\ell_i$ with approximate $\hat{l}_i$.

For the second part we derive a lower bound on the acceptance probability similar to the one from Dereziński et al. [13]. However, while they use an $n \times d_{\mathrm{eff}}(\alpha\mathbf{L})$ Nyström approximation of the matrix $\mathbf{L}$, to avoid direct dependencies on $n$ we are forced to use a less stable approximation $\widehat{\mathbf{L}}$. As a result, controlling $d_{\mathrm{eff}}(\alpha\widehat{\mathbf{L}})$ requires a more careful analysis.

**Beyond uniform subsampling.** One of the implications of our analysis is that more adaptive upper bounds on the marginals $\ell_i$ could further speedup our $\alpha$-DPP sampling approach. In particular, we chose uniform sampling, i.e., a uniform upper bound, for its conceptual simplicity and because knowing an upper bound $\kappa^2$ on the entries of the similarity matrix usually does not require looking at the items, e.g., $\kappa^2$ is always equal to 1 for Gaussian similarity, Cosine similarity or other self normalized similarities. However for other similarities, such as linear similarity, this bound could be very loose. A simple replacement is using the actual diagonal of $\mathbf{L}$, which requires to look at all items and $\mathcal{O}(n)$ time to compute but is usually very scalable. Ideally, one could imagine designing a

sequence of upper bounds starting from cheaper to more computationally expensive, where more advanced techniques such as random projection are used near the end to further filter candidate items.

# 3 Efficient reduction from k-DPP to rescaled DPP via binary search

Given our fast DPP sampler, we can see a $k$-DPP as a sampling process where we first sample $S \sim \text{DPP}(\alpha \mathbf{L})$, check if the sample size $|S|$ is equal to $k$, and then accept or reject the sample accordingly. Rescaling $\mathbf{L}$ by a constant factor $\alpha$ only changes the expected size $d_{\text{eff}}(\alpha \mathbf{L})$ (and not the $k$-DPP), with $\alpha > 1$ increasing the expected size and $\alpha < 1$ decreasing it. Thus, it is natural to imagine that there exists some $\alpha_\star$ for which the acceptance probability is high. Indeed this was recently proven to be possible. Dereziński et al. [13] show that if the *mode* $m_{\alpha_\star}$ of $S_{\alpha_\star} \sim \text{DPP}(\alpha_\star \mathbf{L})$ is equal to $k$, then we will accept with probability at least $\Omega(1/\sqrt{k})$. They also provide an algorithm to find such an $\alpha_\star$. However, this algorithm has a prohibitively high computational cost, $\widetilde{\mathcal{O}}(nk^{10} + k^{15})$, because ensuring that the mode of $\text{DPP}(\alpha_\star \mathbf{L})$ is *exactly $k$* requires an extremely accurate approximation of $\mathbf{L}$. Instead, our approach is to run a binary search to find a good rescaling $\alpha$, which will terminate once the acceptance probability is high enough, regardless of whether $k$ is exactly the mode. Crucially, this binary search only requires a black box $\text{DPP}(\alpha \mathbf{L})$ sampler (such as our $\alpha$-DPP), and it only queries the sampler $\widetilde{\mathcal{O}}(\sqrt{k})$ many times. To prove that the binary search finds a good $\alpha$ in a small number of steps, we establish a new property (Lemma 3) of the Poisson Binomial distribution (the distribution of the subset sizes of $\text{DPP}(\alpha \mathbf{L})$), which should be of independent interest.

## 3.1 Binary search

Our main result in this subsection is Algorithm 2, which requires only oracle access to the samples from a random-size DPP, and finds a rescaling $\hat{\alpha}$ which enables efficient rejection sampling from a $k$-DPP. Note that the provided oracle sampler does not have to be our $\alpha$-DPP sampler, so the algorithm could be paired with other samplers.

**Lemma 1.** *Suppose that we are given an integer $k$, a range $I = [\alpha_{\min}, \alpha_{\max}]$ where $\alpha_{\max} = \gamma \alpha_{\min}$, and access to an oracle which, for any $\alpha \in I$, returns $S \sim \text{DPP}(\alpha \mathbf{L})$. If there exists $\alpha_\star \in I$ such that $k$ is the mode of $|S|$ for $S \sim \text{DPP}(\alpha_\star \mathbf{L})$, then using $O\big(\sqrt{k}\log^2(k\log(\gamma)/\delta)\big)$ calls to the oracle we can find $\hat{\alpha} \in I$ such that with probability $1 - \delta$ we have*

$$\Pr(|S| = k) = \Omega\big(\tfrac{1}{\sqrt{k}}\big), \qquad for \quad S \sim \text{DPP}(\hat{\alpha}\mathbf{L}).$$

The distribution of subset size $|S|$ for $S \sim \text{DPP}(\mathbf{L})$ can be defined via the eigenvalues $\lambda_1 \geq \lambda_2 \geq ...$ of $\mathbf{L}$ (see [25]): if we let $b_i \sim \text{Bernoulli}(\frac{\lambda_i}{\lambda_i+1})$ for $i \geq 1$, then $\sum_i b_i$ is distributed identically to $|S|$. This distribution is known as the *Poisson Binomial*, and it has been extensively studied in the probability literature [38]. The recent result of [13] on the probability of the mode of a Poisson Binomial shows that it is possible to find $\hat{\alpha}$ satisfying the condition of Lemma 1.

**Lemma 2.** *There is an absolute constant $0 < c < 1$ such that for any Poisson Binomial distribution $p : \mathbb{Z}_{\geq 0} \to \mathbb{R}_{\geq 0}$, with mode $k^*$ we have $p(k^*) \geq \frac{c}{\sqrt{k^*+1}}$.*

This result, however, does not provide an efficient way of finding an $\hat{\alpha}$ such that the mode of the subset size distribution of $\text{DPP}(\hat{\alpha}\mathbf{L})$ is $k$. We circumvent this problem by performing a binary search (Algorithm 2) that looks for such an $\hat{\alpha}$, but stops early when it reaches a sufficiently good candidate, avoiding excess computations. To make this rigorous, we establish the following new property of the Poisson Binomial distribution, which should be of independent interest.

**Lemma 3.** *Let $p : \mathbb{Z}_{\geq 0} \to \mathbb{R}_{\geq 0}$ be a Poisson Binomial distribution, and let $k \geq 1$ satisfy $p(k) < \frac{c}{12\sqrt{3(k+1)}}$, where $c$ comes from Lemma 2. Then, $P_{<k} = \sum_{i<k} p(k)$ and $P_{>k} = \sum_{i>k} p(k)$ satisfy:*

*1. if the mode of $p$ is less than $k$, then $P_{>k} \leq \frac{1}{2} - \frac{c}{12}$;*

*2. if the mode of $p$ is greater than $k$, then $P_{<k} \leq \frac{1}{2} - \frac{c}{12}$.*

Informally, the above result states the following: For any $k$, either its probability under the given Poisson Binomial is at least $\Omega(\frac{1}{\sqrt{k}})$, or this $k$ splits the probability mass into two uneven parts, with the larger one containing the mode. Thus, as long as our candidate $\alpha$ does not yield high acceptance probability for $k$, it is easy to make the branching decision in the binary search by estimating the quantities $P_{>k}$ and $P_{<k}$ simply by repeated sampling from $\text{DPP}(\alpha \mathbf{L})$. Note that if the condition

---

**Algorithm 2** Binary search for initializing the $k$-DPP($\mathbf{L}$) sampler

---

**Input:** $0 < \alpha_{\min} < \alpha_{\max}$, sampling oracle for DPP($\alpha\mathbf{L}$), integer $k$ and constants $C > 0, \delta \in (0,1)$

**Output:** $\hat{\alpha}$ such that DPP($\hat{\alpha}\mathbf{L}$) can be used to efficiently sample $k$-DPP($\mathbf{L}$)

1: **for** $s = \{1, \ldots, \lceil \log(\gamma) \rceil\}$ **do**
2:     **if** $\alpha_{\max}/\alpha_{\min} < (1 + \frac{1}{(k+3)^2})$ **then return** $\hat{\alpha} = \alpha_{\min}$
3:     $\bar{\alpha} \leftarrow \sqrt{\alpha_{\min}\alpha_{\max}}$
4:     Sample $S_1, ..., S_t \overset{\text{i.i.d.}}{\sim} \text{DPP}(\bar{\alpha}\mathbf{L})$    where    $t = C\sqrt{k}\log(s/\delta)$
5:     $\hat{P}_k \leftarrow \frac{1}{t}\sum_{i=1}^t \mathbf{1}_{[|S_i|=k]}$
6:     **if** $\hat{P}_k \geq \frac{1}{2} \cdot \frac{c}{12\sqrt{3(k+1)}}$ **then return** $\hat{\alpha} = \bar{\alpha}$
7:     $(\hat{P}_{<k}, \hat{P}_{>k}) \leftarrow \left(\frac{1}{t}\sum_{i=1}^t \mathbf{1}_{[|S_i|<k]}, \frac{1}{t}\sum_{i=1}^t \mathbf{1}_{[|S_i|>k]}\right)$
8:     **if** $\hat{P}_{<k} > \hat{P}_{>k}$ **then** $(\alpha_{\min}, \alpha_{\max}) = (\bar{\alpha}, \alpha_{\max})$ **else** $(\alpha_{\min}, \alpha_{\max}) = (\alpha_{\min}, \bar{\alpha})$
9: **end for**

---

on $p(k)$ is not satisfied, then performing the branching decision could be very expensive, but our algorithm avoids this possibility. The proof of Lemma 1 (Appendix B) follows from Lemmas 2 and 3.

## 3.2 Constructing the initial interval

To initiate our binary search, we must first find a range of values $[\alpha_{\min}, \alpha_{\max}]$, which contains the desired $\alpha_\star$, and also construct a sampling oracle for DPP($\alpha\mathbf{L}$). The binary search procedure is deliberately presented in a way that is agnostic to how these two steps are accomplished, because a number of existing DPP samplers could be adapted to take advantage of Algorithm 2, including [30, 36, 10, 13]. Our implementation of these two steps is different than these previous approaches in that it takes advantage of the structure of the kernel so that it only has to look at a potentially small fraction of the data points. We achieve this with a modified version of the BLESS algorithm [37].

**Lemma 4.** *W.p.* $1 - \delta$ *BLESS-I runs in time* $\widetilde{\mathcal{O}}\left(\min\{\alpha_{\max}\kappa^2, 1\}nk^6 + k^9\right)$ *and satisfies:*

1. *The interval* $[\alpha_{\min}, \alpha_{\max}]$ *is bounded by* $\frac{1}{4}(k-1)/\text{tr}(\mathbf{L}) \leq \alpha_{\min} \leq \alpha_{\max} \leq 8(k+2)/d_{\text{eff}}(\mathbf{L})$
2. *There is* $\alpha_\star \in [\alpha_{\min}, \alpha_{\max}]$ *for which* $k$ *is the mode of* $|S|$ *where* $S \sim \text{DPP}(\alpha_\star\mathbf{L})$;
3. *The dictionary* $\mathcal{D}^{\alpha_{\max}}$ *satisfies the conditions from Theorem 2 for any* $\alpha \in [\alpha_{\min}, \alpha_{\max}]$.

The first two parts of the lemma ensure that the interval $I = [\alpha_{\min}, \alpha_{\max}]$ is a valid input for the binary search in Algorithm 2 and that its size $\gamma = \alpha_{\max}/\alpha_{\min} \leq 4\text{tr}(\mathbf{L})/d_{\text{eff}}(\mathbf{L})$ is bounded in the log-scale. The last part implies that $\alpha$-DPP can be used by that algorithm as the oracle sampler.

At a high level, Algorithm 6 proceeds by starting with a small $\alpha^0$ that is guaranteed to be a valid lower bound for the interval, and for which a dictionary $\mathcal{D}^0$ can be constructed simply via uniform sampling. Then we repeatedly double the $\alpha$ and refine the dictionary, until we reach $\alpha^i$ such that we can ensure that with high probability $d_{\text{eff}}(\alpha^i\mathbf{L}) \geq k + 1$ which makes it a valid upper bound for the interval (then, this $\alpha^i$ becomes $\alpha_{\max}$).

## 3.3 Overall time complexity of $k$-DPP sampling

Putting together all the results from the previous sections, we can finally bound the computational complexity of our $k$-DPP sampler, which first uses BLESS-I (Algorithm 6) to construct a dictionary and search interval, and then applies the binary search of (Algorithm 2) using our $\alpha$-DPP sampler (Algorithm 1) as the sampling oracle. Once again note that in the following computational analysis we will use conservative values for many parameters, notably $r$ from $\alpha$-DPP and $q$ from BLESS-I, as they are suggested from the theory. However in practice it is always better to start from a more optimistic value, and keep doubling them only if the sampler repeatedly fails to accept. Importantly, samples generated this way will still be exactly distributed according to the DPP, as all the approximations used in our approach only influence the runtime of our algorithm, and not the correctness of its acceptance, which always holds.

By Lemma 4, the preprocessing step of running BLESS-I takes $\widetilde{\mathcal{O}}\left(\min\{\alpha_{\max}\kappa^2,1\}nk^6 + k^9\right)$ and generates a dictionary $\mathcal{D}$ with size $\widetilde{\mathcal{O}}(k^3)$. Since $d_{\text{eff}}(\alpha\mathbf{L}) \leq d_{\text{eff}}(\alpha_{\max}\mathbf{L}) \leq \mathcal{O}(k)$ for all $\alpha$ in the search interval, each call to the $\alpha$-DPP sampler also requires at most $\widetilde{\mathcal{O}}\left(\min\{\alpha_{\max}\kappa^2 k,1\}nk^6 + k^9\right)$. Finally, the binary search invokes $\alpha$-DPP at most $\widetilde{\mathcal{O}}(\sqrt{k})$ times so the overall runtime is $\widetilde{\mathcal{O}}\left((\min\{\alpha_{\max}\kappa^2 k,1\}nk^6 + k^9)\cdot\sqrt{k}\right)$. We now provide a bound on $\alpha_{\max}$.

**Lemma 5.** *For any matrix $\mathbf{L}$ and $0 < \alpha \leq 1$, we have $d_{\text{eff}}(\alpha\mathbf{L})/d_{\text{eff}}(\mathbf{L}) \geq \alpha \geq d_{\text{eff}}(\alpha\mathbf{L})/\text{tr}(\mathbf{L})$.*

Applied to $\alpha_{\max}$, we obtain $\alpha_{\max} \leq d_{\text{eff}}(\alpha_{\max}\mathbf{L})/d_{\text{eff}}(\mathbf{L}) \leq \mathcal{O}(k/d_{\text{eff}}(\mathbf{L}))$, giving us the final runtime of $\widetilde{\mathcal{O}}\left((\min\{k^2\kappa^2/d_{\text{eff}}(\mathbf{L}),1\}nk^6 + k^9)\cdot\sqrt{k}\right)$ reported in Theorem 1.

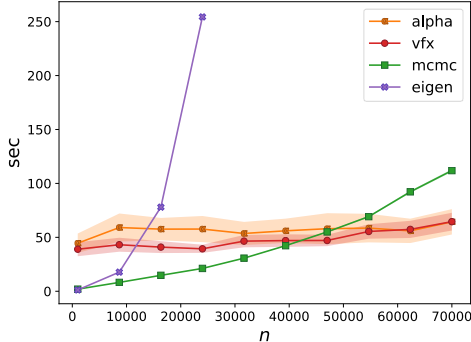

Figure 1: Small scale experiment

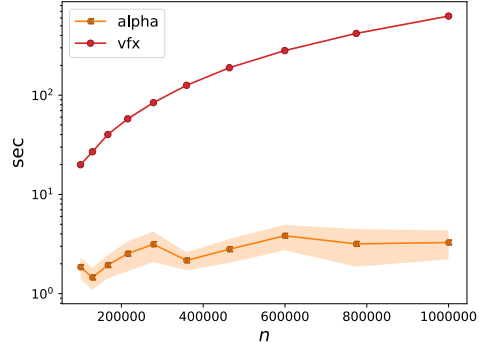

Figure 2: Large scale experiment.

## 4 Experiments

In this section, we evaluate our $\alpha$-DPP sampler on a benchmark[2] introduced by [13] (see Appendix D). The benchmark uses subsets of the infinite MNIST dataset [33] with $d = 784$ and $n$ varying up to $10^6$. All experiments are executed on a 28 core Xeon E5-2680 v4. Each experiment is repeated multiple times, and we report mean values and a 95% confidence interval.

**Baselines:** we compare $\alpha$-DPP with DPP-VFX [13], an MCMC sampler [3] and a sampler based on eigendecompositions [25, 22]. All algorithms are implemented in `python` as part[3] of DPPy [20]. Due to their similar input, we use the same oversampling parameters (see Appendix D) for $\alpha$-DPP and DPP-VFX. We run the MCMC sampler for $\mathcal{O}(nk)$ iterations to guarantee mixing [3]. For more details on hyperparameter tuning we refer to Appendix D.

**Results** We begin by reporting results on a smaller subset of data (Figure 1) where even the non-efficient samplers can be run. We use an `rbf` similarity with $\sigma = \sqrt{3d}$, and set $k = 10$ to match the number of digit classes in MNIST. Note that for $n = 70000$ BLESS-I estimates $d_{\text{eff}}(\mathbf{L}) \approx 300$, validating our assumption of $d_{\text{eff}}(\mathbf{L}) \gg k$. Thanks to this mismatch, we can see how $\alpha$-DPP maintains a *constant* runtime as $n$ grows, and increasingly matches or outpaces competing baselines as $n$ grows. In particular, it becomes faster than the eigendecomposition based sampler (which cannot scale beyond $n = 24000$) or the MCMC sampler. However, the gap is still sufficiently small that DPP-VFX, the previously fastest $k$-DPP sampler available, remains competitive. Note that, unlike the MCMC and eigendecomposition based samplers, $\alpha$-DPP and DPP-VFX sample from a $k$-DPP by repeatedly sampling from a random-size DPP until they generate a sample with size exactly $k$. While our theory ensures that this will happen after only a small number of rejections, this creates some overhead cost relative to the other two methods, which is noticeable for small values of $n$.

For larger datasets we consider only the scalable samplers, $\alpha$-DPP and DPP-VFX. We consider again an `rbf` similarity, but this time we choose $n$ up to $10^6$ and $\sigma = \sqrt{10}$. This further increases the gap between $k$ and $d_{\text{eff}}(\mathbf{L})$, with BLESS-I estimating $d_{\text{eff}}(\mathbf{L}) \approx 1000$. We report results in Figure 2, with runtime shown in log-scale. In this regime, the gap between DPP-VFX and $\alpha$-DPP widens, as DPP-VFX cannot use rescaling to reduce the final dictionary size from $d_{\text{eff}}(\mathbf{L})$ to $d_{\text{eff}}(\hat{\alpha}\mathbf{L}) \approx k$, and has to compute $n$ marginal probabilities since it does not leverage uniform intermediate subsampling. In particular, thanks to the uniform sampling step, we see that $\alpha$-DPP's runtime does not grow as $n$

grows, since all the expensive computations are performed in the small intermediate subset which is hardly sensitive to $n$. We note that, due to using a smaller dictionary, $\alpha$-DPP requires about 2-5x more trials in the rejection sampling step, which leads to larger variance in the runtime.

In Figure 3, we report the fraction of data that is observed by $\alpha$-DPP in the large scale experiment. This quantity, denoted as $\beta$ in Theorem 1, is responsible for much of the computational gains of the algorithm over DPP-VFX, reported in Figure 2. Note that the remaining $1 - \beta$ portion of the data does not ever need to be loaded into the program's memory, which leads to a significant reduction in memory accesses. We observe that as the data size increases, the fraction of items observed by $\alpha$-DPP goes down to as little as 1% for $n = 10^6$, which is why the runtime of $\alpha$-DPP stays roughly flat, whereas the runtime of DPP-VFX grows.

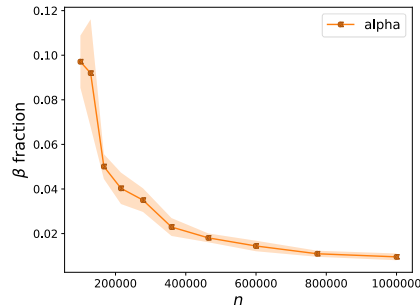

Figure 3: Fraction of items observed by $\alpha$-DPP.

## Broader impact

DPPs were discovered in the 70s by Odile Macchi to model repulsion of particle distributions in fermions, so improvements in samplers may help in modelling physical simulations. In bringing faster DPP samplers to machine learning we aim to enable a better handling of diversity through this rigorous theoretical framework.

## Acknowledgments and Disclosure of Funding

MD thanks the NSF for funding via the NSF TRIPODS program.

## Footnotes

[2] https://github.com/LCSL/dpp-vfx

[3] Our implementation of $\alpha$-DPP is included in the supplementary material, and it is also available in DPPy.

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
