[Supplementary Material]

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

[4]Note that we are defining the kernel $K$ as a function on indices, but since we focus on DPPs defined on PSD matrices, everything can be immediately extended to any standard PSD kernel $K(\cdot, \cdot) : \mathcal{X} \times \mathcal{X} \to \mathbb{R}$ defined on an arbitrary input space $\mathcal{X}$.

[5]Again we focus on a feature map from indices to a finite dimensional space. All the results can be immediately extended to a feature map $\varphi(\cdot) : \mathcal{X} \to \mathcal{H}$ that maps from an arbitrary input space into a reproducing kernel Hilbert space, e.g., Gaussian kernel and Gaussian feature maps. Notice that in our setting given a PSD matrix $\mathbf{L}$ the eigenspace of $\mathbf{L}$ suffices to construct an appropriate feature map $\varphi(\cdot) : [n] \to \mathbb{R}^n$ with $D = n$. In particular, we have an explicit expression for $\varphi(\cdot)$ based on the eigendecomposition $\mathbf{L} = \mathbf{U}\Sigma\mathbf{U}^\top$ of $\mathbf{L}$. Since $\mathbf{L}$ is psd, $\Sigma$ is a diagonal matrix with non-negative entries, and we can define $\Sigma^{+/2}$ as the square root of its pseudo-inverse. Then the feature map becomes $\varphi(\cdot) \stackrel{def}{=} \Sigma^{+/2}\mathbf{U}^\top K([n], \cdot)$. A similar argument can be made using the Cholesky decomposition of $\mathbf{L}$.

[6]Or an identity operator on an RKHS in general

[7]Following DPPy's API, these hyperparameters are denoted as `rls_oversample_bless` and `rls_oversample_dppvfx` in our code.

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

# A  Correctness and efficiency of $\alpha$-DPP (Algorithm 1)

In this section we prove the theorems stated in Sections 2 and 3 claiming the correctness and efficiency of DPP-VFX. In particular, we split Theorem 2 into two parts.

**Lemma 6.** *Given any psd matrix* $\mathbf{L}$*, dictionary* $\mathcal{D}$*, positive weights* $\mathbf{W}$*,* $r \geq 1$*, and positive* $\alpha > 0$*,* $\alpha$*-DPP returns* $S \sim \mathrm{DPP}(\alpha\mathbf{L})$*.*

**Lemma 7.** *If* $r \geq d_{\mathrm{eff}}(\alpha\mathbf{L}) \geq 1/2$*,* $\mathcal{D}$ *and* $\mathbf{W}$ *are* $(1/d_{\mathrm{eff}}(\alpha\mathbf{L}), \alpha)$*-accurate,* $\mathcal{D}$ *satisfies* $|\mathcal{D}| \leq 10 d_{\mathrm{eff}}(\alpha\mathbf{L})$*, and* $d_{\mathrm{eff}}(\alpha\widehat{\mathbf{L}}) \leq 10 d_{\mathrm{eff}}(\alpha\mathbf{L})$*, then with probability* $1 - \delta$*,* $\alpha$*-DPP (Algorithm 1) runs in time*

$$\mathcal{O}\Big( \big[ \min\{\alpha\kappa^2 d_{\mathrm{eff}}(\alpha\mathbf{L}), 1\} \cdot n \cdot d_{\mathrm{eff}}(\alpha\mathbf{L})^6 \log^2(n/\delta) + d_{\mathrm{eff}}(\alpha\mathbf{L})^9 \log^3(n/\delta) \big] \cdot \log(1/\delta) \Big).$$

## A.1  Notation

We start by introducing some additional notation. First, let us describe the so-called kernel-based view of DPPs. We associate with our similarity matrix $\mathbf{L}$ a similarity function or a kernel[4] function $K(\cdot, \cdot) : [n] \times [n] \to \mathbb{R}$ such that $K(i, j)$ is equal to the $(i, j)$-th entry of $\mathbf{L}$.

We also generalize the notation just defined in a way that given multi-sets $A$ and $B$, $K(A, B) \stackrel{def}{=} \mathbf{L}_{A,B}$ returns the matrix containing the corresponding rows and columns of $\mathbf{L}$. Note that if $A$ or $B$ contains duplicates (e.g., the $i$-th index appears twice in $A$) the matrix $K(A, B)$ will consequently contain duplicate rows and columns. Finally, note that in this notation the original matrix can be written as $\mathbf{L} = K([n], [n])$.

We also denote with $\varphi(\cdot) : [n] \to \mathbb{R}^D$ the so-called feature map associated with $\mathbf{L}$ and $K(\cdot, \cdot)$ such that $K(i, j) = \varphi(i)^\top \varphi(j)$, where $D$ can be arbitrarily large or infinite.[5] Just as with $K$, we also extend $\varphi(\cdot)$ to operate on multi-set, such that given $A = \{i_1, \ldots, i_m\}$ (potentially with duplicates $i_j = i_l$), we have $\varphi(A) = [\varphi(i_1), \ldots, \varphi(i_m)]^\top \in \mathbb{R}^{m \times D}$.

Using the above notation, we have $\mathbf{L} = K([n], [n]) = \varphi([n])\varphi([n])^\top$. Note also that the corresponding operator $\varphi([n])^\top \varphi([n])$ can be decomposed as a sum of outer products $\varphi([n])^\top \varphi([n]) = \sum_{i=1}^n \varphi(i)\varphi(i)^\top$.

We also use the following notation to indicate common sampling distributions:

- $u \sim \mathrm{Poisson}(\lambda)$ as a non-negative integer sampled from a Poisson distribution with intensity $0 < \lambda$;

- $\rho \sim \mathrm{Uniform}(u, [n])$ as a set of size $u$ sampled uniformly i.i.d. with replacement from $[n]$; i.e., $\rho = (\rho_1, \ldots, \rho_u) \stackrel{\text{i.i.d.}}{\sim} (1/n, \ldots, 1/n)$.

- $z \sim \mathrm{Bernoulli}(p)$ as the $\{0, 1\}$ r.v. sampled from a Bernoulli distribution w.p. $0 \leq p \leq 1$;

- $s \sim \mathrm{Binomial}(k, p)$ as the non-negative integer in the range $[0, k]$ sampled from a Binomial distribution with $0 \leq k$ Bernoulli repetitions each with probability $0 \leq p \leq 1$

- $(s_1, \ldots, s_n) \sim \mathrm{GenBinomial}(k, \{p_i\}_{i=1}^n)$ as the vector of positive integers $[0, k]^n$ with $0 \leq k$ sampled according to $\mathbf{P}\left((s_1, \ldots, s_n)\right) = \frac{n!}{\prod_{i=1}^n s_i!} \prod_{i=1}^n p_i^{s_i}$ such that $\sum_{i=1}^n s_i = k$.

- $\sigma \sim \mathrm{Multinomial}(k, \{p_i\}_{i=1}^n)$ as a set of size $k$ sampled i.i.d. with replacement from $[n]$ according to probabilities $0 \leq p_i \leq 1$ with $\sum_{i=1}^n p_i = 1$, i.e., $\sigma = (\sigma_1, \ldots, \sigma_k) \stackrel{\text{i.i.d.}}{\sim} (p_1, \ldots, p_n)$.

## A.2 Proof of Lemma 6 (exact sampling)

To prove that $\alpha$-DPP is an exact sampler we show that $\sigma$ is sampled according to an appropriate R-DPP, and that therefore we can invoke Proposition 1.

*Proof of Lemma 6.* Given the approximate marginals $l_i$ from Equation 1, let us denote with $\widetilde{d}_{\text{eff}}(\alpha\mathbf{L}) \overset{def}{=} \sum_{i=1}^{n} l_i$ their sum, or approximate effective dimension. Note that Algorithm 1 *never* computes $\widetilde{d}_{\text{eff}}(\alpha\mathbf{L})$ explicitly, nor does it compute all approximate marginals $l_i$. Nonetheless, our first claim is that the inner loop of $\alpha$-DPP is proposing a candidate $\sigma$ sampled according to the approximate marginals $l_i$ even without computing them all.

**Lemma 8.** *The set $\sigma$ generated by Algorithm 1 before Line 11 is distributed as*

$$\sigma = \{\sigma_1, \ldots, \sigma_t\} \overset{\text{i.i.d.}}{\sim} (l_1/\widetilde{d}_{\text{eff}}(\alpha\mathbf{L}), \ldots, l_n/\widetilde{d}_{\text{eff}}(\alpha\mathbf{L})), \qquad t \sim Poisson\left(re^{1/r}\widetilde{d}_{\text{eff}}(\alpha\mathbf{L})\right).$$

Then, we show that the rejection sampling step of Line 11 is valid.

**Lemma 9.** *Given any psd matrix $\mathbf{L}$, dictionary $\mathcal{D}$, positive weights $\mathbf{W}$, $r \geq 1$, and positive $\alpha > 0$, the acceptance probability $\frac{e^{d_{\text{eff}}(\alpha\widehat{\mathbf{L}})}\det(\mathbf{I}+\alpha\widetilde{\mathbf{L}}_\sigma)}{e^{t/r}\det(\mathbf{I}+\alpha\widehat{\mathbf{L}})} \leq 1$ is valid.*

Let $\widetilde{\sigma}$ denote the random variable distributed as $\sigma$ is after exiting the repeat loop. Combining Lemma 9 with the fact that $t \sim \text{Poisson}(re^{1/r}\widetilde{d}_{\text{eff}}(\alpha\mathbf{L}))$ is a Poisson r.v. it follows that

$$\begin{aligned}
\Pr(\widetilde{\sigma} \in A) &\propto \mathbb{E}_\sigma\left[\mathbf{1}_{[\sigma \in A]}\frac{e^{d_{\text{eff}}(\alpha\widehat{\mathbf{L}})}\det(\mathbf{I}+\widetilde{\mathbf{L}}_\sigma)}{e^{t/r}\det(\mathbf{I}+\widehat{\mathbf{L}})}\right] \\
&\propto \sum_{t=0}^{\infty} \frac{(r\,e^{1/r}\widetilde{d}_{\text{eff}}(\alpha\mathbf{L}))^t}{e^{r\,e^{1/r}\widetilde{d}_{\text{eff}}(\alpha\mathbf{L})}\,t!} \cdot e^{-t/r}\,\mathbb{E}_\sigma\left[\mathbf{1}_{[\sigma \in A]}\det(\mathbf{I}+\widetilde{\mathbf{L}}_\sigma) \mid t\right] \\
&\propto \mathbb{E}_{t'}\left[\mathbb{E}_\sigma\left[\mathbf{1}_{[\sigma \in A]}\det(\mathbf{I}+\widetilde{\mathbf{L}}_\sigma) \mid t = t'\right]\right] \quad \text{for } t' \sim \text{Poisson}(r\widetilde{d}_{\text{eff}}(\alpha\mathbf{L})),
\end{aligned}$$

which matches the numerator of Definition 1 for a R-DPP$^{r\widetilde{d}_{\text{eff}}(\alpha\mathbf{L})}_{\{l_i/\widetilde{d}_{\text{eff}}(\alpha\mathbf{L})\}_{i=1}^n}$. All that remains is to show that the distribution integrates properly, i.e., the denominator also matches. We do this by generalizing a determinantal equality to our modified reweighting.

**Proposition 2** ([10]). *If $t \sim Poisson(r\widetilde{d}_{\text{eff}}(\alpha\mathbf{L}))$ and $(\sigma_1, \ldots, \sigma_t) \overset{\text{i.i.d.}}{\sim} \left(\frac{l_1}{\widetilde{d}_{\text{eff}}(\alpha\mathbf{L})}, \ldots, \frac{l_n}{\widetilde{d}_{\text{eff}}(\alpha\mathbf{L})}\right)$ then*

$$\mathbb{E}_{t,\sigma}\left[\det\left(\mathbf{I}+\widetilde{\mathbf{L}}_\sigma\right)\right] = \det\left(\mathbf{I}+\mathbf{L}\right).$$

This shows that $\widetilde{\sigma} \sim \text{R-DPP}^{r\widetilde{d}_{\text{eff}}(\alpha\mathbf{L})}_{\{l_i/\widetilde{d}_{\text{eff}}(\alpha\mathbf{L})\}_{i=1}^n}$. The claim follows from Proposition 1. ∎

*Proof of Lemma 8.* Before starting, we will use two well known connections of the Poisson distribution with GenBinomial and Binomial r.v. [27]. The first useful Poisson property is that for any set of positive weights $\{\lambda_i\}_{i=1}^n$ the random variables $X_1 \sim \text{Poisson}(\lambda_1), \ldots, X_n \sim \text{Poisson}(\lambda_n)$ and the random variables

$$k \sim \text{Poisson}\left(\sum_{i=1}^n \lambda_i\right), \qquad \{X_1, \ldots, X_n\}|k \sim \text{GenBinomial}\left(k, \frac{\lambda_i}{\sum_{i=1}^n \lambda_i}\right)$$

are equally distributed. Note that for this identity to hold we do not need to explicitly compute $\sum_{i=1}^n \lambda_i$, as we can simply sample $n$ Poisson r.v.-s and obtain the normalization effect in the GenBinomial sample for free using the conditioning on $k$. The second useful Poisson property we will use is that if $k \sim \text{Poisson}(\lambda)$ and $Y|k \sim \text{Binomial}(k, p)$ then $Y \sim \text{Poisson}(\lambda \cdot p)$.

---

**Algorithm 3** $\alpha$-DPP reformulation in terms of $u_i$ and $s_i$

---

**Input:** $\alpha$, $\mathbf{L}$, an $m$-element dictionary $\mathcal{D}$, weight matrix $\mathbf{W} \in \mathbb{R}^{m \times m}$, $r \geq 1$
 1: Set $\widehat{\mathbf{L}} = \mathbf{W}^{1/2} \mathbf{L}_{\mathcal{D},\mathcal{D}} \mathbf{W}^{1/2} \in \mathbb{R}^{m \times m}$
 2: **repeat**
 3:     **for** $i = \{1, \ldots, n\}$ **do**
 4:        Sample $u_i \sim \text{Poisson}(re^{1/r} \alpha \kappa^2)$
 5:        **if** $u_i > 0$ **then**
 6:           Compute $l_i$ using Equation 1
 7:           Sample $s_i \sim \text{Binomial}(u_i, l_i/(\alpha\kappa^2))$
 8:        **else**
 9:           Set $s_i = 0$
10:        **end if**
11:        Add $s_i$ copies of $i$ to $\sigma$
12:     **end for**
13:     Set $t = |\sigma|$, $[\widetilde{\mathbf{L}}_\sigma]_{ij} = \frac{1}{r\sqrt{l_{\sigma_i} l_{\sigma_j}}} [\mathbf{L}]_{\sigma_i \sigma_j}$
14:     Sample $Acc \sim \text{Bernoulli}\left( e^{d_{\text{eff}}(\alpha\widehat{\mathbf{L}}) - t/r} \det(\mathbf{I} + \alpha\widetilde{\mathbf{L}}_\sigma)/\det(\mathbf{I} + \alpha\widehat{\mathbf{L}}) \right)$
15: **until** $Acc = \text{true}$
16: Sample $\widetilde{S} \sim \text{DPP}(\alpha\widetilde{\mathbf{L}}_\sigma)$
17: **return** $S = \{\sigma_i : i \in \widetilde{S}\}$

---

Let us denote with $u_i = \sum_{j=1}^u \mathbb{I}\{\rho_j = i\}$ the multiplicity of index $i$ in $\rho$, such that we have a set of $n$ random variables $\{u_i\}_{i=1}^n$ and that $u = \sum_{i=1}^n u_i$. Then, from the previous relationship, we can instantly see that sampling $u_i \sim \text{Poisson}(re^{1/r}b)$ is equivalent to sampling

$$u \sim \text{Poisson}\Big( \sum_{i=1}^n re^{1/r} \alpha\kappa^2 \Big) = \text{Poisson}(re^{1/r} n\alpha\kappa^2),$$

$$\{u_1, \ldots, u_n\} | u \sim \text{GenBinomial}\left( u, \frac{re^{1/r}\alpha\kappa^2}{re^{1/r}n\alpha\kappa^2} \right) = \text{GenBinomial}\left( u, \left\{ \tfrac{1}{n} \right\} \right) \cdot$$

We can now connect $u_i$ and $\rho$. In particular sampling $\rho | u \sim \text{Uniform}(u, [n])$ is equivalent to sampling $\{u_1, \ldots, u_n\} | u \overset{\text{i.i.d.}}{\sim} \text{GenBinomial}\left( u, \left\{ \tfrac{1}{n} \right\} \right)$ and then adding $u_i$ copies of $i$ to $\rho | u$ for each $i \in [n]$.

Starting from this characterization, let us now denote with $s_i = \sum_{j=1}^t \mathbb{I}\{\sigma_j = i\}$ the multiplicity of index $i$ in $\sigma$, such that we have a set of $n$ random variables $\{s_i\}_{i=1}^n$ and that $t = \sum_{i=1}^n s_i$. We can now formally describe Line 7 of Algorithm 1 as a binomial sampling step: first we sample $u_i \sim \text{Poisson}(re^{1/r}\alpha\kappa^2)$, and then we sample $s_i | u_i \sim \text{Binomial}(u_i, l_i/(\alpha\kappa^2))$. To see this, we can just sum over all $z_j$ that correspond to the $i$-th element, of which we have exactly $u_i$, and remember that a sum of i.i.d. Bernoulli is a Binomial. We also have to take care of the fact that the Binomial probability is well defined, i.e., smaller than 1, but it is easy to see that $l_i \leq \alpha[\mathbf{L}]_{i,i} \leq \alpha\kappa^2$ and $l_i/(\alpha\kappa^2) \leq 1$. We can now use the second fact about Poissons, namely that sampling $u_i \sim \text{Poisson}(re^{1/r}b)$ and then $s_i | u_i \sim \text{Binomial}(u_i, \frac{l_i}{b})$ is equivalent to sampling $s_i \sim \text{Poisson}(re^{1/r}b \cdot \frac{l_i}{b}) = \text{Poisson}(re^{1/r}l_i)$.

Finally we can once again use the equivalence between Poisson and GenBinomial sampling to see that sampling $s_i \sim \text{Poisson}(re^{1/r}l_i)$ for each $i \in [n]$ is equivalent to sampling

$$t \sim \text{Poisson}\Big( \sum_{i=1}^n ql_i \Big) = \text{Poisson}(re^{1/r}\widetilde{d}_{\text{eff}}(\alpha\mathbf{L})), \quad \{s_1, \ldots, s_n\} | t \sim \text{GenBinomial}\left( t, \frac{l_i}{\widetilde{d}_{\text{eff}}(\alpha\mathbf{L})} \right),$$

which in turn implies that by adding $s_i$ copies of the index $i$ to $\sigma$, which is what Algorithm 1 is doing, we are sampling according to

$$\sigma = \{\sigma_1, \ldots, \sigma_t\} \overset{\text{i.i.d.}}{\sim} (l_1/\widetilde{d}_{\text{eff}}(\alpha\mathbf{L}), \ldots, l_n/\widetilde{d}_{\text{eff}}(\alpha\mathbf{L})), \qquad t \sim \text{Poisson}(re^{1/r}\widetilde{d}_{\text{eff}}(\alpha\mathbf{L})),$$

without ever explicitly computing $\widetilde{d}_{\text{eff}}(\alpha\mathbf{L})$.

For completeness, we also include the two implicit reformulations of Algorithm 1 that we just described as Algorithm 3 and Algorithm 4. Note that all three algorithms are strictly equivalent, but depending on the actual implementation they have different complexities. For example, Algorithm 4

---

**Algorithm 4** $\alpha$-DPP reformulation in terms of $s_i$

---

**Input:** $\alpha$, $\mathbf{L}$, an $m$-element dictionary $\mathcal{D}$, weight matrix $\mathbf{W} \in \mathbb{R}^{m \times m}$, $r \geq 1$

1: **repeat**
2:    **for** $i = \{1, \ldots, n\}$ **do**
3:       Compute $l_i$ using Equation 1
4:       Sample $s_i \sim \text{Poisson}(re^{1/r}l_i)$
5:       Add $s_i$ copies of $i$ to $\sigma$
6:    **end for**
7:    Set $t = |\sigma|$, $[\widetilde{\mathbf{L}}_\sigma]_{ij} = \frac{1}{r\sqrt{l_{\sigma_i}l_{\sigma_j}}}[\mathbf{L}]_{\sigma_i\sigma_j}$
8:    Sample $Acc \sim \text{Bernoulli}\left(e^{d_{\text{eff}}(\alpha\widehat{\mathbf{L}})-t/r}\det(\mathbf{I}+\alpha\widetilde{\mathbf{L}}_\sigma)/\det(\mathbf{I}+\alpha\widehat{\mathbf{L}})\right)$
9: **until** $Acc = $ true
10: Sample $\widetilde{S} \sim \text{DPP}(\alpha\widetilde{\mathbf{L}}_\sigma)$
11: **return** $S = \{\sigma_i : i \in \widetilde{S}\}$

---

needs to compute all marginals in advance. We chose to include Algorithm 1 in the main paper as the version that more clearly highlights the uniform sampling step. ∎

*Proof of Lemma 9.* The first reason we introduced the kernel-based DPP notation is to be able to succinctly use Sylvester's identity to equate determinants in the matrix and feature view of the DPP, i.e.,

$$\det(\mathbf{I} + \mathbf{L}) = \det(\mathbf{I} + \varphi([n])\varphi([n])^\top) = \det(\mathbf{I} + \varphi([n])^\top\varphi([n])) = \det\left(\mathbf{I} + \sum_{i=1}^n \varphi(i)^\top\varphi(i)\right),$$

where the size of the identity matrix[6] $\mathbf{I}$ is either $n$ or $D$ and it is clear from the context. Similarly, the denominator $\det(\mathbf{I} + \alpha\widehat{\mathbf{L}}) = \det(\mathbf{I} + \alpha\mathbf{W}^{1/2}\mathbf{L}_{\mathcal{D},\mathcal{D}}\mathbf{W}^{1/2})$ in the rejection loop becomes

$$\det(\mathbf{I} + \alpha\mathbf{W}^{1/2}\mathbf{L}_{\mathcal{D},\mathcal{D}}\mathbf{W}^{1/2}) = \det(\mathbf{I} + \alpha\mathbf{W}^{1/2}\varphi(\mathcal{D})\varphi(\mathcal{D})^\top\mathbf{W}^{1/2}) = \det(\mathbf{I} + \alpha\varphi(\mathcal{D})^\top\mathbf{W}\varphi(\mathcal{D})).$$

Finally, given $\sigma$ let us denote with $\widetilde{\varphi}(i) = \frac{1}{\sqrt{rl_i}}\varphi(i)$ a rescaled feature map, where once again we extend $\widetilde{\varphi}(\sigma) = \text{Diag}(\frac{1}{\sqrt{rl_{\sigma_i}}})_{i=1}^m\varphi(\sigma)$ to multi-sets. Then the numerator in the rejection loop becomes

$$\det(\mathbf{I} + \alpha\widetilde{\mathbf{L}}_\sigma) = \det(\mathbf{I} + \alpha\widetilde{\varphi}(\sigma)\widetilde{\varphi}(\sigma)^\top)$$
$$= \det(\mathbf{I} + \alpha\widetilde{\varphi}(\sigma)^\top\widetilde{\varphi}(\sigma)) = \det\left(\mathbf{I} + \alpha\sum_{j=1}^t \frac{1}{rl_{\sigma_j}}\varphi(\sigma_j)\varphi(\sigma_j)^\top\right).$$

The second reason we introduce this notation is that the formulation of the approximate marginals $l_i$ is much simplified and becomes (see [5, 37] for details)

$$l_i = \alpha[\mathbf{L} - \alpha\mathbf{L}_{[n],\mathcal{D}}^\top(\alpha\mathbf{L}_{\mathcal{D},\mathcal{D}} + \mathbf{W}^{-1})^{-1}\mathbf{L}_{[n],\mathcal{D}}]_{i,i} = \alpha\varphi(i)^\top(\mathbf{I} + \alpha\varphi(\mathcal{D})^\top\mathbf{W}\varphi(\mathcal{D}))^{-1}\varphi(i). \quad (2)$$

Using the kernel-based view of DPPs and the reformulation of most quantities, we can now move from characterizing the distribution of $\sigma$, to computing the final acceptance probability $\mathbf{P}(Acc|\sigma)$. In particular, to guarantee correctness we must guarantee that the rejection step is valid, i.e., that the acceptance probability is bounded by 1. For this we rewrite the acceptance condition as

$$\frac{\det(\mathbf{I} + \alpha\widetilde{\mathbf{L}}_\sigma)}{\det(\mathbf{I} + \alpha\widehat{\mathbf{L}})} = \frac{\det(\mathbf{I} + \alpha\widetilde{\varphi}(\sigma)^\top\widetilde{\varphi}(\sigma))}{\det(\mathbf{I} + \alpha\varphi(\mathcal{D})^\top\mathbf{W}\varphi(\mathcal{D}))}.$$

Similarly to [13], we can use the inequality $\det(\mathbf{I} + \mathbf{A}) \leq \exp\{\text{tr}(\mathbf{A})\}$, which follows immediately by applying the bound $1 + x \leq e^x$ to each singular value of $\mathbf{A}$. We obtain

$$\frac{\det(\mathbf{I} + \alpha\widetilde{\varphi}(\sigma)^\top\widetilde{\varphi}(\sigma))}{\det(\mathbf{I} + \alpha\varphi(\mathcal{D})^\top\mathbf{W}\varphi(\mathcal{D}))}$$
$$= \det\left((\mathbf{I} + \alpha\widetilde{\varphi}(\sigma)^\top\widetilde{\varphi}(\sigma))(\mathbf{I} + \alpha\varphi(\mathcal{D})^\top\mathbf{W}\varphi(\mathcal{D}))^{-1}\right)$$
$$= \det\left(\mathbf{I} + (\alpha\widetilde{\varphi}(\sigma)^\top\widetilde{\varphi}(\sigma) - \alpha\varphi(\mathcal{D})^\top\mathbf{W}\varphi(\mathcal{D}))(\mathbf{I} + \alpha\varphi(\mathcal{D})^\top\mathbf{W}\varphi(\mathcal{D}))^{-1}\right)$$
$$\leq \exp\left\{\text{tr}\left((\alpha\widetilde{\varphi}(\sigma)^\top\widetilde{\varphi}(\sigma) - \alpha\varphi(\mathcal{D})^\top\mathbf{W}\varphi(\mathcal{D}))(\mathbf{I} + \alpha\varphi(\mathcal{D})^\top\mathbf{W}\varphi(\mathcal{D}))^{-1}\right)\right\}$$
$$= \exp\{\underbrace{\text{tr}\left(\alpha\widetilde{\varphi}(\sigma)^\top\widetilde{\varphi}(\sigma)(\mathbf{I} + \alpha\varphi(\mathcal{D})^\top\mathbf{W}\varphi(\mathcal{D}))^{-1}\right)}_{(a)} - \underbrace{\text{tr}\left(\alpha\varphi(\mathcal{D})^\top\mathbf{W}\varphi(\mathcal{D})(\mathbf{I} + \alpha\varphi(\mathcal{D})^\top\mathbf{W}\varphi(\mathcal{D}))^{-1}\right)}_{(b)}\}.$$

For $(b)$, we can see that by definition $\operatorname{tr}\left(\alpha\varphi(\mathcal{D})^\top\mathbf{W}\varphi(\mathcal{D})(\mathbf{I}+\alpha\varphi(\mathcal{D})^\top\mathbf{W}\varphi(\mathcal{D}))^{-1}\right)=d_{\text{eff}}(\alpha\widehat{\mathbf{L}})$.
For $(a)$, we have

$$(a)=\operatorname{tr}\left(\alpha\widetilde{\varphi}(\sigma)^\top\widetilde{\varphi}(\sigma)(\mathbf{I}+\alpha\varphi(\mathcal{D})^\top\mathbf{W}\varphi(\mathcal{D}))^{-1}\right)=\operatorname{tr}\left(\alpha\widetilde{\varphi}(\sigma)(\mathbf{I}+\alpha\varphi(\mathcal{D})^\top\mathbf{W}\varphi(\mathcal{D}))^{-1}\widetilde{\varphi}(\sigma)^\top\right)$$

$$=\sum_{j=1}^t\alpha\widetilde{\varphi}(\sigma_j)^\top(\mathbf{I}+\alpha\varphi(\mathcal{D})^\top\mathbf{W}\varphi(\mathcal{D}))^{-1}\widetilde{\varphi}(\sigma_j)=\sum_{j=1}^t\frac{1}{rl_{\sigma_j}}\alpha\varphi(\sigma_j)^\top(\mathbf{I}+\alpha\varphi(\mathcal{D})^\top\mathbf{W}\varphi(\mathcal{D}))^{-1}\varphi(\sigma_j)$$

$$=\sum_{j=1}^t\frac{1}{rl_{\sigma_j}}l_{\sigma_j}=\frac{t}{r}.$$

Putting $(a)$ and $(b)$ together we have

$$\mathbf{P}(Acc=\text{true}|\sigma)=\frac{\det(\mathbf{I}+\alpha\widetilde{\varphi}(\sigma)^\top\widetilde{\varphi}(\sigma))}{\det(\mathbf{I}+\alpha\varphi(\mathcal{D})^\top\mathbf{W}\varphi(\mathcal{D}))}\cdot\exp\left\{d_{\text{eff}}(\alpha\widehat{\mathbf{L}})-\frac{t}{r}\right\}$$

$$\leq\exp\left\{\frac{t}{r}-d_{\text{eff}}(\alpha\widehat{\mathbf{L}})\right\}\cdot\exp\left\{d_{\text{eff}}(\alpha\widehat{\mathbf{L}})-\frac{t}{r}\right\}=e^0=1.$$

$\blacksquare$

*Proof of Proposition 2.* We first rewrite the equality as

$$\mathbb{E}_{t,\sigma}\left[\det\left(\mathbf{I}+\sum_{j=1}^t\frac{1}{rl_{\sigma_j}}\varphi(\sigma_j)\varphi(\sigma_j)^\top\right)\right]=\det\left(\mathbf{I}+\sum_{i=1}^n\varphi(i)\varphi(i)^\top\right).$$

Dereziński [10] showed the following identity when sampling $t\sim\text{Poisson}(r)$ and then sampling a multi-set $\sigma$ with $t$ elements i.i.d. from any arbitrary distribution,

$$\mathbb{E}_{t,\sigma}\left[\det\left(\mathbf{I}+\varphi(\sigma)^\top\varphi(\sigma)\right)\right]=\det\left(\mathbf{I}+r\mathbb{E}_{\sigma_1}\left[\varphi(\sigma_1)\varphi(\sigma_1)^\top\right]\right).$$

Applying this to our $t\sim\text{Poisson}(r\widetilde{d}_{\text{eff}}(\alpha\mathbf{L}))$ and the distribution of $\sigma$ we have

$$\mathbb{E}_{t,\sigma}\left[\det\left(\mathbf{I}+\sum_{j=1}^t\frac{1}{rl_{\sigma_j}}\varphi(\sigma_j)\varphi(\sigma_j)^\top\right)\right]=\det\left(\mathbf{I}+r\widetilde{d}_{\text{eff}}(\alpha\mathbf{L})\mathbb{E}_{\sigma_1}\left[\varphi(\sigma_1)\varphi(\sigma_1)^\top\right]\right)$$

$$=\det\left(\mathbf{I}+r\widetilde{d}_{\text{eff}}(\alpha\mathbf{L})\sum_{i=1}^n\frac{l_i}{\widetilde{d}_{\text{eff}}(\alpha\mathbf{L})}\frac{1}{rl_i}\varphi(\sigma_i)\varphi(\sigma_i)^\top\right)$$

$$=\det\left(\mathbf{I}+\sum_{i=1}^n\varphi(i)\varphi(i)^\top\right).$$

$\blacksquare$

### A.3 Proof of Lemma 7 (efficiency)

*Proof of Lemma 7.* We need to lower bound the acceptance probability $\mathbf{P}(Acc|\sigma)$. Note that this is equivalent to lower bounding $\mathbb{E}[Acc=\text{true}]$ since it is a $\{0,1\}$ random variable.

$$\mathbf{P}(Acc=\text{true})=\mathbb{E}_{\sigma,t}\left[\frac{e^{d_{\text{eff}}(\alpha\widehat{\mathbf{L}})}\det(\mathbf{I}+\alpha\widetilde{\varphi}(\sigma)^\top\widetilde{\varphi}(\sigma))}{e^{t/r}\det(\mathbf{I}+\alpha\widehat{\mathbf{L}})}\right]=\frac{e^{d_{\text{eff}}(\alpha\widehat{\mathbf{L}})}}{\det(\mathbf{I}+\alpha\widehat{\mathbf{L}})}\mathbb{E}_{\sigma,t}\left[\frac{\det(\mathbf{I}+\alpha\widetilde{\varphi}(\sigma)^\top\widetilde{\varphi}(\sigma))}{e^{t/r}}\right]$$

$$=\frac{e^{d_{\text{eff}}(\alpha\widehat{\mathbf{L}})}}{\det(\mathbf{I}+\alpha\widehat{\mathbf{L}})}\cdot\sum_{t=0}^\infty\mathbb{E}_\sigma\left[\det(\mathbf{I}+\alpha\widetilde{\varphi}(\sigma)^\top\widetilde{\varphi}(\sigma))|\,t\,\right]\frac{1}{e^{t/r}}\frac{\left(re^{1/r}\widetilde{d}_{\text{eff}}(\alpha\mathbf{L})\right)^t}{t!\cdot e^{re^{1/r}\widetilde{d}_{\text{eff}}(\alpha\mathbf{L})}},$$

where we expanded the expectation with respect to $t\sim\text{Poisson}(re^{1/r}\widetilde{d}_{\text{eff}}(\alpha\mathbf{L}))$. Focusing on the last term we have

$$\frac{1}{e^{t/r}}\frac{\left(re^{1/r}\widetilde{d}_{\text{eff}}(\alpha\mathbf{L})\right)^t}{t!\cdot e^{re^{1/r}\widetilde{d}_{\text{eff}}(\alpha\mathbf{L})}}\cdot=\frac{1}{e^{t/r}}\frac{r^te^{t/r}\widetilde{d}_{\text{eff}}(\alpha\mathbf{L})^t}{t!\cdot e^{re^{1/r}\widetilde{d}_{\text{eff}}(\alpha\mathbf{L})}}=\frac{r^t\widetilde{d}_{\text{eff}}(\alpha\mathbf{L})^t}{t!\cdot e^{re^{1/r}\widetilde{d}_{\text{eff}}(\alpha\mathbf{L})}}=\frac{(r\widetilde{d}_{\text{eff}}(\alpha\mathbf{L}))^t}{t!\cdot e^{r\widetilde{d}_{\text{eff}}(\alpha\mathbf{L})}}\frac{e^{r\widetilde{d}_{\text{eff}}(\alpha\mathbf{L})}}{e^{re^{1/r}\widetilde{d}_{\text{eff}}(\alpha\mathbf{L})}}.$$

Recognizing that $\frac{(r\widetilde{d}_{\text{eff}}(\alpha\mathbf{L}))^t}{t! \cdot e^{r\widetilde{d}_{\text{eff}}(\alpha\mathbf{L})}}$ is the density of a $t \sim \text{Poisson}(r\widetilde{d}_{\text{eff}}(\alpha\mathbf{L}))$, we can apply Proposition 2,

$$\mathbf{P}(Acc = \text{true}) = \frac{e^{d_{\text{eff}}(\alpha\widehat{\mathbf{L}})}}{\det(\mathbf{I} + \alpha\widehat{\mathbf{L}})} \cdot \sum_{t=0}^{\infty} \mathbb{E}_\sigma \big[ \det(\mathbf{I} + \alpha\widetilde{\varphi}(\sigma)^\top\widetilde{\varphi}(\sigma))| \, t \, \big] \frac{1}{e^{t/r}} \frac{(re^{1/r}\widetilde{d}_{\text{eff}}(\alpha\mathbf{L}))^t}{t! \cdot e^{re^{1/r}\widetilde{d}_{\text{eff}}(\alpha\mathbf{L})}}$$

$$= \frac{e^{d_{\text{eff}}(\alpha\widehat{\mathbf{L}})}}{\det(\mathbf{I} + \alpha\widehat{\mathbf{L}})} \frac{e^{r\widetilde{d}_{\text{eff}}(\alpha\mathbf{L})}}{e^{re^{1/r}\widetilde{d}_{\text{eff}}(\alpha\mathbf{L})}} \cdot \sum_{t=0}^{\infty} \mathbb{E}_\sigma \big[ \det(\mathbf{I} + \alpha\widetilde{\varphi}(\sigma)^\top\widetilde{\varphi}(\sigma))| \, t \, \big] \frac{(r\widetilde{d}_{\text{eff}}(\alpha\mathbf{L}))^t}{t! \cdot e^{r\widetilde{d}_{\text{eff}}(\alpha\mathbf{L})}}$$

$$= \frac{e^{d_{\text{eff}}(\alpha\widehat{\mathbf{L}})}}{\det(\mathbf{I} + \alpha\widehat{\mathbf{L}})} \frac{e^{r\widetilde{d}_{\text{eff}}(\alpha\mathbf{L})}}{e^{re^{1/r}\widetilde{d}_{\text{eff}}(\alpha\mathbf{L})}} \cdot \det(\mathbf{I} + \alpha\varphi([n])^\top\varphi([n]))$$

$$= \frac{e^{d_{\text{eff}}(\alpha\widehat{\mathbf{L}})} e^{r\widetilde{d}_{\text{eff}}(\alpha\mathbf{L})}}{e^{re^{1/r}\widetilde{d}_{\text{eff}}(\alpha\mathbf{L})}} \frac{\det(\mathbf{I} + \alpha\varphi([n])^\top\varphi([n]))}{\det(\mathbf{I} + \alpha\varphi(\mathcal{D})^\top\mathbf{W}\varphi(\mathcal{D}))}.$$

To lower bound this quantity we will again upper bound the inverse $\frac{\det(\mathbf{I}+\alpha\varphi(\mathcal{D})^\top\mathbf{W}\varphi(\mathcal{D}))}{\det(\mathbf{I}+\alpha\varphi([n])^\top\varphi([n]))}$ as follows

$$\frac{\det(\mathbf{I}+\alpha\varphi(\mathcal{D})^\top\mathbf{W}\varphi(\mathcal{D}))}{\det(\mathbf{I}+\alpha\varphi([n])^\top\varphi([n]))} \leq \exp\big\{ \text{tr}\big( (\alpha\varphi(\mathcal{D})^\top\mathbf{W}\varphi(\mathcal{D}) - \alpha\varphi([n])^\top\varphi([n]))(\mathbf{I} + \alpha\varphi([n])^\top\varphi([n]))^{-1} \big) \big\}$$

$$= \exp\big\{ \text{tr}\big( \alpha\varphi(\mathcal{D})^\top\mathbf{W}\varphi(\mathcal{D})(\mathbf{I} + \alpha\varphi([n])^\top\varphi([n]))^{-1} \big) - d_{\text{eff}}(\alpha\mathbf{L}) \big\}.$$

Inverting the relationship and putting it all together we have

$$\mathbf{P}(Acc_\sigma = \text{true})$$

$$\geq \exp\Big\{ d_{\text{eff}}(\alpha\widehat{\mathbf{L}}) + r\widetilde{d}_{\text{eff}}(\alpha\mathbf{L}) - re^{1/r}\widetilde{d}_{\text{eff}}(\alpha\mathbf{L}) + d_{\text{eff}}(\alpha\mathbf{L}) - \text{tr}\big( \alpha\varphi(\mathcal{D})^\top\mathbf{W}\varphi(\mathcal{D})(\mathbf{I} + \alpha\varphi([n])^\top\varphi([n]))^{-1} \big) \Big\}.$$

Using the bound $e^{1/r} \leq 1 + 1/r + 1/r^2$ for $r \geq 1$ we simplify

$$r\widetilde{d}_{\text{eff}}(\alpha\mathbf{L}) - re^{1/r}\widetilde{d}_{\text{eff}}(\alpha\mathbf{L}) \geq r\widetilde{d}_{\text{eff}}(\alpha\mathbf{L}) - r\widetilde{d}_{\text{eff}}(\alpha\mathbf{L}) - \widetilde{d}_{\text{eff}}(\alpha\mathbf{L}) - \widetilde{d}_{\text{eff}}(\alpha\mathbf{L})/r \geq -\widetilde{d}_{\text{eff}}(\alpha\mathbf{L}) - \widetilde{d}_{\text{eff}}(\alpha\mathbf{L})/r$$

and obtain the final

$$\mathbf{P}(Acc_\sigma = \text{true})$$

$$\geq \exp\Big\{ d_{\text{eff}}(\alpha\widehat{\mathbf{L}}) - \widetilde{d}_{\text{eff}}(\alpha\mathbf{L}) + d_{\text{eff}}(\alpha\mathbf{L}) - \text{tr}\big( \alpha\varphi(\mathcal{D})^\top\mathbf{W}\varphi(\mathcal{D})(\mathbf{I} + \alpha\varphi([n])^\top\varphi([n]))^{-1} \big) - \widetilde{d}_{\text{eff}}(\alpha\mathbf{L})/r \Big\}.$$

Using the definition of $(\varepsilon, \alpha)$-accuracy, we have

$$\widetilde{d}_{\text{eff}}(\alpha\mathbf{L}) = \sum_{i=1}^{n} \alpha\varphi(i)^\top(\mathbf{I} + \alpha\varphi(\mathcal{D})^\top\mathbf{W}\varphi(\mathcal{D}))^{-1}\varphi(i)$$

$$= \text{tr}\big( \alpha\varphi([n])(\mathbf{I} + \alpha\varphi(\mathcal{D})^\top\mathbf{W}\varphi(\mathcal{D}))^{-1}\varphi([n])^\top \big)$$

$$\leq \tfrac{1}{1-\varepsilon}\text{tr}\big( \alpha\varphi([n])(\mathbf{I} + \alpha\varphi([n])^\top\varphi([n]))^{-1}\varphi([n])^\top \big)$$

$$= \tfrac{1}{1-\varepsilon}d_{\text{eff}}(\alpha\mathbf{L}) = (1 + \tfrac{\varepsilon}{1-\varepsilon})d_{\text{eff}}(\alpha\mathbf{L}),$$

and therefore $-\widetilde{d}_{\text{eff}}(\alpha\mathbf{L}) + d_{\text{eff}}(\alpha\mathbf{L}) \geq \tfrac{\varepsilon}{1-\varepsilon}d_{\text{eff}}(\alpha\mathbf{L})$. On the other side

$$\text{tr}\big( \alpha\varphi(\mathcal{D})^\top\mathbf{W}\varphi(\mathcal{D})(\mathbf{I} + \alpha\varphi([n])^\top\varphi([n]))^{-1} \big) \leq \tfrac{1}{1-\varepsilon}\text{tr}\big( \alpha\varphi(\mathcal{D})^\top\mathbf{W}\varphi(\mathcal{D})(\mathbf{I} + \alpha\varphi(\mathcal{D})^\top\mathbf{W}\varphi(\mathcal{D}))^{-1} \big)$$

$$= \tfrac{1}{1-\varepsilon}d_{\text{eff}}(\alpha\widehat{\mathbf{L}}) = (1 + \tfrac{\varepsilon}{1-\varepsilon})d_{\text{eff}}(\alpha\widehat{\mathbf{L}}),$$

and therefore $\widetilde{d}_{\text{eff}}(\alpha\widehat{\mathbf{L}}) - \text{tr}\big( \alpha\varphi(\mathcal{D})^\top\mathbf{W}\varphi(\mathcal{D})(\mathbf{I} + \alpha\varphi([n])^\top\varphi([n]))^{-1} \big) \geq \tfrac{\varepsilon}{1-\varepsilon}d_{\text{eff}}(\alpha\widehat{\mathbf{L}})$. Putting it all together, we obtain our result $\mathbf{P}(Acc_\sigma = \text{true}) \geq \exp\{-(\varepsilon(d_{\text{eff}}(\alpha\mathbf{L}) + d_{\text{eff}}(\alpha\widehat{\mathbf{L}})) + \widetilde{d}_{\text{eff}}(\alpha\mathbf{L})/r))\}$.
To bound $\varepsilon d_{\text{eff}}(\alpha\mathbf{L})$ we simply use the fact that the dictionary is $1/d_{\text{eff}}(\alpha\mathbf{L})$ accurate. Secondly to bound $\widetilde{d}_{\text{eff}}(\alpha\mathbf{L})/r$ we use the fact that by Equation 2 and Proposition 5

$$\widetilde{d}_{\text{eff}}(\alpha\mathbf{L}) = \sum_{i=1}^{n} l_i = \sum_{i=1}^{n} \alpha\varphi(i)^\top(\mathbf{I} + \alpha\varphi(\mathcal{D})^\top\mathbf{W}\varphi(\mathcal{D}))^{-1}\varphi(i)$$

$$\leq \frac{1}{1-\varepsilon}\sum_{i=1}^{n} \alpha\varphi(i)^\top(\mathbf{I} + \alpha\varphi([n])^\top\varphi([n]))^{-1}\varphi(i) = \frac{1}{1-\varepsilon}\sum_{i=1}^{n} \ell_i(\alpha\mathbf{L}) = \frac{d_{\text{eff}}(\alpha\mathbf{L})}{1-\varepsilon}.$$

Combining this with the fact that $\varepsilon \leq 1/2$ and that $r \geq d_{\text{eff}}(\alpha\mathbf{L})$ we have that $\widetilde{d}_{\text{eff}}(\alpha\mathbf{L})/r \leq 2$.

Finally, to bound $\varepsilon d_{\text{eff}}(\alpha\widehat{\mathbf{L}})$, first we bound

$$
\begin{aligned}
d_{\text{eff}}(\alpha\widehat{\mathbf{L}}) &= \text{tr}(\alpha\mathbf{W}^{1/2}\mathbf{L}_{\mathcal{D}}\mathbf{W}^{1/2}(\alpha\mathbf{W}^{1/2}\mathbf{L}_{\mathcal{D}}\mathbf{W}^{1/2} + \mathbf{I})^{-1}) \\
&= \text{tr}(\alpha\mathbf{W}^{1/2}\varphi(\mathcal{D})\varphi(\mathcal{D})^{\top}\mathbf{W}^{1/2}(\alpha\mathbf{W}^{1/2}\varphi(\mathcal{D})\varphi(\mathcal{D})^{\top}\mathbf{W}^{1/2} + \mathbf{I})^{-1}) \\
&= \text{tr}(\alpha\varphi(\mathcal{D})^{\top}\mathbf{W}^{1/2}\mathbf{W}^{1/2}\varphi(\mathcal{D})^{\top}(\alpha\varphi(\mathcal{D})^{\top}\mathbf{W}^{1/2}\mathbf{W}^{1/2}\varphi(\mathcal{D})^{\top} + \mathbf{I})^{-1}) \\
&= \text{tr}(\alpha\mathbf{W}\varphi(\mathcal{D})^{\top}(\alpha\varphi(\mathcal{D})^{\top}\mathbf{W}\varphi(\mathcal{D})^{\top} + \mathbf{I})^{-1}\varphi(\mathcal{D})^{\top}) \\
&= \sum_{j=1}^{m}[\mathbf{W}]_{j,j}\alpha\varphi(\mathcal{D}_j)^{\top}(\mathbf{I} + \alpha\varphi(\mathcal{D})^{\top}\mathbf{W}\varphi(\mathcal{D}))^{-1}\varphi(\mathcal{D}_j) \\
&\leq \sum_{j=1}^{m}[\mathbf{W}]_{j,j}\tfrac{1}{1-\varepsilon}\alpha\varphi(\mathcal{D}_j)^{\top}(\mathbf{I} + \alpha\varphi([n])^{\top}\varphi([n]))^{-1}\varphi(\mathcal{D}_j) = \sum_{j=1}^{m}[\mathbf{W}]_{j,j}\tfrac{1}{1-\varepsilon}\ell_{\mathcal{D}_j}(\alpha\mathbf{L})
\end{aligned}
$$

where the last inequality used again Equation 2 and Proposition 5. To continue we have to use the following result for BLESS, the specific dictionary construction algorithm used by $\alpha$-DPP, which follows immediately from Proposition 4 in Appendix C.

**Proposition 3.** *For some $\alpha' \geq \alpha$, let $\mathcal{D}$ be a dictionary generated using* BLESS-I *ran with parameter $q \geq 54\kappa^2\frac{(2\varepsilon+1)^2}{\varepsilon^2}\log(12n^2/\delta)$ and $\varepsilon \leq \min\{1/2, 1/d_{\text{eff}}(\alpha'\mathbf{L})\}$. Then w.p. $1 - \delta$*

- *the dictionary and weights are $(\varepsilon, \alpha')$-accurate,*

- *the weights $\mathbf{W}$ obtained satisfy $[\mathbf{W}]_{j,j} \leq \max\{\frac{1}{1-\varepsilon}\frac{1}{q\ell_{\mathcal{D}_j}(\alpha'\mathbf{L})}, 1\}$,*

- *the size of the dictionary $m = |\mathcal{D}|$ is bounded as $m/q \leq 2d_{\text{eff}}(\alpha'\mathbf{L})$.*

Applying this to the previous bound, and using the $(1/d_{\text{eff}}(\alpha\mathbf{L}), \alpha)$-accuracy, $\varepsilon \leq 1/2$ and the fact that $\ell_{\mathcal{D}_j}(\alpha\mathbf{L}) \leq \ell_{\mathcal{D}_j}(\alpha'\mathbf{L})$ for $\alpha' \geq \alpha$ we obtain

$$
\begin{aligned}
\sum_{j=1}^{m}[\mathbf{W}]_{j,j}\tfrac{1}{1-\varepsilon}\ell_j(\alpha\mathbf{L}) &\leq \sum_{j=1}^{m}\max\{\tfrac{d_{\text{eff}}(\alpha'\mathbf{L})}{d_{\text{eff}}(\alpha'\mathbf{L})-1}\tfrac{1}{d_{\text{eff}}(\alpha'\mathbf{L})^2\ell_{\mathcal{D}_j}(\alpha'\mathbf{L})}, 1\}2\ell_{\mathcal{D}_j}(\alpha\mathbf{L}) \\
&= 2\sum_{j=1}^{m}\max\{\tfrac{d_{\text{eff}}(\alpha'\mathbf{L})}{d_{\text{eff}}(\alpha'\mathbf{L})-1}\tfrac{1}{d_{\text{eff}}(\alpha'\mathbf{L})^2}\tfrac{\ell_{\mathcal{D}_j}(\alpha\mathbf{L})}{\ell_{\mathcal{D}_j}(\alpha'\mathbf{L})}, \ell_{\mathcal{D}_j}(\alpha\mathbf{L})\} \\
&\leq 2\sum_{j=1}^{m}\max\{\tfrac{d_{\text{eff}}(\alpha'\mathbf{L})}{d_{\text{eff}}(\alpha'\mathbf{L})-1}\tfrac{1}{d_{\text{eff}}(\alpha'\mathbf{L})^2}, \ell_{\mathcal{D}_j}(\alpha\mathbf{L})\} \\
&\leq 2\sum_{j=1}^{m}\left(\tfrac{d_{\text{eff}}(\alpha'\mathbf{L})}{d_{\text{eff}}(\alpha'\mathbf{L})-1}\tfrac{1}{d_{\text{eff}}(\alpha'\mathbf{L})^2} + \ell_{\mathcal{D}_j}(\alpha\mathbf{L})\right).
\end{aligned}
$$

To conclude, we have that since BLESS does not include duplicates in $\mathcal{D}$,

$$
\sum_{j=1}^{m}\ell_{\mathcal{D}_j}(\alpha\mathbf{L}) \leq \sum_{i=1}^{n}\ell_i(\alpha\mathbf{L}) = d_{\text{eff}}(\alpha\mathbf{L}).
$$

Now using the second result from Proposition 3 on $m$ we have

$$
\sum_{j=1}^{m}\tfrac{d_{\text{eff}}(\alpha'\mathbf{L})}{d_{\text{eff}}(\alpha'\mathbf{L})-1}\tfrac{1}{d_{\text{eff}}(\alpha'\mathbf{L})^2} = m\tfrac{d_{\text{eff}}(\alpha'\mathbf{L})}{d_{\text{eff}}(\alpha'\mathbf{L})-1}\tfrac{1}{d_{\text{eff}}(\alpha'\mathbf{L})^2} \leq 2d_{\text{eff}}(\alpha'\mathbf{L})\tfrac{d_{\text{eff}}(\alpha'\mathbf{L})}{d_{\text{eff}}(\alpha'\mathbf{L})-1}\tfrac{1}{d_{\text{eff}}(\alpha'\mathbf{L})^2} \leq 2.
$$

$\blacksquare$

# B  Proofs for the binary search algorithm

In this section we present omitted proofs for the binary search algorithm. The key properties of a Poisson Binomial which we will use are summarized in the following two lemmas.

**Lemma 10.** *Let $p : \mathbb{Z}_{\geq 0} \to \mathbb{R}_{\geq 0}$ be a Poisson Binomial distribution. Then:*

1. *$p$ is unimodal, i.e., if $k^*$ is the mode of $p$, then $p(1) \leq ... \leq p(k^*) \geq p(k^* + 1) \geq ...$;*

2. *$p$ is log-concave, i.e., $log(p)$ is a concave function over the support of $p$;*

3. *the median of $p$ is one of $k^* - 1$, $k^*$ and $k^* + 1$.*

**Lemma 11** ([9])**.** *Given a Poisson Binomial with mean $\bar{k}$ and mode $k^*$, let $k \overset{def}{=} \lfloor \bar{k} \rfloor$. Then:*

$$k^* = \begin{cases} k & if \quad k \leq \bar{k} < k + \frac{1}{k+2}, \\ k \ \ or \ \ k + 1 & if \quad k + \frac{1}{k+2} \leq \bar{k} \leq k + 1 - \frac{1}{n-k+1}, \\ k + 1 & if \quad k + 1 - \frac{1}{n-k+1} < \bar{k} \leq k + 1. \end{cases}$$

Note that these statements are independent of how we break ties in the definitions of the mode and the median, but for the sake of concreteness, suppose that we round down when choosing between a pair of (consecutive) mode/median candidates.

**Lemma 3** (restated)**.** *Let $p : \mathbb{Z}_{\geq 0} \to \mathbb{R}_{\geq 0}$ be a Poisson Binomial distribution, and let $k \geq 1$ satisfy $p(k) < \frac{c}{12\sqrt{3(k+1)}}$, where $c$ comes from Lemma 2. Then, $P_{<k} = \sum_{i<k} p(k)$ and $P_{>k} = \sum_{i>k} p(k)$ satisfy:*

1. *if the mode of $p$ is less than $k$, then $P_{>k} \leq \frac{1}{2} - \frac{c}{12}$;*
2. *if the mode of $p$ is greater than $k$, then $P_{<k} \leq \frac{1}{2} - \frac{c}{12}$.*

*Proof of Lemma 3.* Let $k^*$ be the mode of $p$ and let $p^*$ denote $p(k^*)$. From Lemma 10 it follows that $k \neq k^*$. Suppose that $k^* > k$ (which implies that $k^* \geq 2$) and define:

$$t \overset{def}{=} \min \left\{ i \in \{1, ..., k^*\} \quad \text{subject to} \quad p(k^* - i) < \frac{p^*}{(1 + \beta p^*)^i} \right\},$$

where $\beta = 2 + c/2.5$ is chosen so that the following inequalities (used later) hold: (a) $\frac{1}{\beta} \leq \frac{1}{2} - \frac{c}{12}$, (b) $\frac{1}{(1+\beta)^2} \geq \frac{1}{12}$ and (c) $e^\beta \leq 12$. If no $i$ exists satisfying the above constraint, then we let $t = k^* + 1$ and use $p(-1) = 0$ for convenience. We consider two cases.

**Case 1:** $t \leq \lceil c/p^* \rceil + 1$. Since $p^* \geq \frac{c}{\sqrt{k^*+1}}$, it follows that $t \leq \lceil \sqrt{k^*+1} \rceil + 1$. Note that if $k > k^* - t$ then $k + 1 \geq k^* + 1 - \lceil \sqrt{k^*+1} \rceil \geq (k^* + 1)/3$ and $p(k) \geq p^*(1 + \beta p^*)^{-1/p^*} > \frac{c}{e^\beta \sqrt{k^*+1}} \geq \frac{c}{12\sqrt{3(k+1)}}$ which is a contradiction, so we must have $k \leq k^* - t$. Furthermore, using the definition of $t$ as well as unimodality and log-concavity of $p$, for any $i \geq t$ we have:

$$\frac{p(k^* - i + 1)}{p(k^* - i)} \geq \frac{p(k^* - t + 1)}{p(k^* - t)} \geq 1 + \beta p^*.$$

Thus, $p(k^* - i) < \frac{p^*}{(1+\beta p^*)^i}$ for all $i \geq t$ and it follows that:

$$P_{<k} \leq \sum_{i>t} p(k^* - i) \leq \frac{p^*}{(1 + \beta p^*)^t} \sum_{i \geq 1} \frac{1}{(1 + \beta p^*)^i} \leq \frac{p^*}{(1 + \beta p^*)^t} \frac{1}{\beta p^*} \leq \frac{1}{\beta} \leq \frac{1}{2} - \frac{c}{12}.$$

**Case 2:** $t > \lceil c/p^* \rceil + 1$. This implies that for any $i \leq \lceil c/p^* \rceil$ we have $p(k^* - i) \geq p^*(1 + \beta p^*)^{-c/p^*} > \frac{c}{12\sqrt{3(k+1)}} > p(k)$ so $k \leq k^* - \lceil c/p^* \rceil - 1$. Note that the median of $p$ is no less than $k^* - 1$ so:

$$P_{<k} \leq \underbrace{\sum_{i<k^*-1} p(i)}_{\leq 1/2} - \underbrace{\sum_{i=2}^{\lceil c/p^* \rceil + 1} p(k^* - i)}_{B}.$$

If $p^* \geq \frac{c}{2\beta}$, then it suffices to note that

$$B \geq p(k^* - 2) \geq \frac{p^*}{(1 + \beta p^*)^2} \geq \min\left\{\frac{c/(2\beta)}{(1 + c/2)^2}, \frac{1}{(1 + \beta)^2}\right\} \geq \frac{c}{12},$$

whereas if $p^* < \frac{c}{2\beta}$, then, we have:

$$B = p^*\left(\frac{1 - (1 + \beta p^*)^{-\lceil c/p^* \rceil - 1}}{1 - (1 + \beta p^*)^{-1}} - 1 - (1 + \beta p^*)^{-1}\right) = \frac{1}{\beta}\left(1 - (1 + \beta p^*)^{-\lceil c/p^* \rceil}\right) - \frac{p^*}{1 + \beta p^*}$$

$$\geq \frac{1}{\beta}\left(1 - 2^{-\beta c} - c/2\right) \geq \frac{1}{\beta}\left(1 - (1 - \beta c/3) - c/2\right) = \frac{c}{3} - \frac{c}{2\beta} > \frac{c}{12},$$

which completes the proof when $k^* > k$, and the case of $k^* < k$ follows analogously. ∎

We are now ready to establish the correctness of the binary search procedure that is used to prove Lemma 1, with pseudo-code given in Algorithm 2. In the proof we will use the following standard form of the Chernoff bound.

**Lemma 12** (Chernoff bound). *Let $X_1, ..., X_t$ be independent Bernoulli variables and let $\bar{X} = \frac{1}{t}\sum_i X_i$. Then, for any $0 < \epsilon \leq 1$, we have:*

$$\Pr\left(|\bar{X} - \mathbb{E}[\bar{X}]| \geq \epsilon \cdot \mathbb{E}[\bar{X}]\right) \leq 2e^{-\epsilon^2 t \mathbb{E}[\bar{X}]/3}.$$

**Lemma 1** (restated). *Suppose that we are given an integer $k$, a range $I = [\alpha_{\min}, \alpha_{\max}]$ where $\alpha_{\max} = \gamma \alpha_{\min}$, and access to an oracle which, for any $\alpha \in I$, returns $S \sim \mathrm{DPP}(\alpha \mathbf{L})$. If there exists $\alpha_\star \in I$ such that $k$ is the mode of $|S|$ for $S \sim \mathrm{DPP}(\alpha_\star \mathbf{L})$, then using $O\big(\sqrt{k}\log^2(k\log(\gamma)/\delta)\big)$ calls to the oracle we can find $\hat{\alpha} \in I$ such that with probability $1 - \delta$ we have*

$$\Pr(|S| = k) = \Omega\big(\tfrac{1}{\sqrt{k}}\big), \qquad \text{for} \quad S \sim \mathrm{DPP}(\hat{\alpha}\mathbf{L}).$$

*Proof of Lemma 1.* Let $\mathrm{PB}(\alpha \mathbf{L})$ denote the size distribution of $\mathrm{DPP}(\alpha \mathbf{L})$. Since the binary search is performed in the log-scale, it takes at most $O(\log(k\log(\gamma)))$ steps to reduce the interval ratio $\frac{\alpha_{\max}}{\alpha_{\min}}$ from $\gamma$ to $1 + \frac{1}{(k+3)^2}$. We first establish concentration of $\hat{P}_k$ around its mean $\mathbb{E}[\hat{P}_k] = \Pr(|S_1| = k) = p(k)$, where $p$ is the probability function of $\mathrm{PB}(\bar{\alpha}\mathbf{L})$. Define $f(x) = \Pr(\bar{X} \geq q/2)$ where $\bar{X} = \frac{1}{t}\sum_{i=1}^t X_i$ and $X_i$ are drawn i.i.d. from $\mathrm{Bernoulli}(x)$, with $q = \frac{c}{12\sqrt{3(k+1)}}$. Lemma 12 implies that, choosing a sufficiently large constant $C$ in Algorithm 2, we have:

$$\max\left\{f(q/4), 1 - f(q)\right\} \leq 2e^{-tq/12} \leq \frac{\delta}{4s^2},$$

where $s$ is the number of the current branching step. Note that if $p(k) > q$ then $\Pr(\hat{P}_k < q/2) \leq 1 - f(q) \leq \delta/(4s^2)$ whereas if $p(k) < q/4$, then $\Pr(\hat{P}_k \geq q/2) \leq f(q/4) \leq \delta/(4s^2)$, so putting this together we conclude that:

$$\Pr\left(\left(\hat{P}_k \geq \tfrac{q}{2} \Rightarrow p(k) \geq \tfrac{q}{4}\right) \wedge \left(\hat{P}_k < \tfrac{q}{2} \Rightarrow p(k) < q\right)\right)$$

$$= 1 - \Pr\left(\left(\hat{P}_k \geq \tfrac{q}{2} \wedge p(k) < \tfrac{q}{4}\right) \vee \left(\hat{P}_k < \tfrac{q}{2} \wedge p(k) \geq q\right)\right) \geq 1 - \frac{\delta}{4s^2}.$$

Thus, conditioning on the above high probability event ensures that when the **if** statement in Line 6 of Algorithm 2 succeeds then $\hat{\alpha}$ satisfies the condition from Lemma 1 because $p(k) \geq q/4 = \Omega(\frac{1}{\sqrt{k}})$, and when the if statement fails, then the assumption of Lemma 3 is satisfied because $p(k) < q$.

We now move on to the branching step of the binary search (Line 8). Our assumptions ensure that the initial interval $(\alpha_{\min}, \alpha_{\max})$ contains an $\alpha_\star$ such that $k$ is the mode of $\mathrm{PB}(\alpha_\star \mathbf{L})$. Our goal is to show that the branching step preserves this invariant throughout the procedure. As discussed above, when entering the branching step, with high probability we have $p(k) < q$, so that we can use Lemma 3. Note that $\mathbb{E}[\hat{P}_{<k}] = P_{<k}$ and $\mathbb{E}[\hat{P}_{>k}] = P_{>k}$, as defined in the lemma, and the goal of the branching statement is to determine whether $P_{<k} > P_{<k}$, since that tells us on which side of $k$ is the

mode of $\mathrm{PB}(\bar{\alpha}K)$. Conditioned on a high probability event, we know that either $P_{<k} \leq \frac{1}{2} - \frac{c}{12}$ or $P_{>k} \leq \frac{1}{2} - \frac{c}{12}$. Suppose the former holds. Then, we have:

$$P_{>k} = 1 - (P_{<k} + p(k)) \geq 1 - (\tfrac{1}{2} - \tfrac{c}{12} + \tfrac{c}{12\sqrt{3}}) \geq \tfrac{1}{2} + \tfrac{c}{30},$$

and an analogous bound follows for $P_{<k}$ in the latter case. If $P_{>k} \geq \frac{1}{2} + \frac{c}{30}$ (call it event $E$), then we can once again apply Lemma 12 to show that (for a sufficiently large constant $C$),

$$\Pr\left(\hat{P}_{<k} > \hat{P}_{>k} \mid E\right) \geq \Pr\left(\hat{P}_{<k} \geq \tfrac{1}{2} \mid E\right) \geq \Pr\left(|\hat{P}_{<k} - \mathbb{E}[P_{<k}]| < \tfrac{c}{30} \mid E\right)$$
$$\geq 1 - 2\exp\left\{-(\tfrac{c}{30})^2 \tfrac{1}{2} t/3\right\} \geq 1 - \delta/(4s^2),$$

and an analogous claim follows when $P_{>k} \leq \frac{1}{2} - \frac{c}{12}$. Conditioning on this high probability event implies (via Lemma 3) that the interval constructed after branching still satisfies the invariant. A union bound implies that the probability that any of the events we have conditioned on fails (throughout the algorithm) is bounded by $\sum_{s \geq 1} \frac{2\delta}{4s^2} \leq \delta$. Thus, with probability $1 - \delta$ the last interval used in the search will still satisfy the invariant. It remains to show that when the **if** statement in Line 2 succeeds then either $\alpha_{\min}$ or $\alpha_{\max}$ satisfies the claim of Lemma 1. To that end, since $k \geq \lfloor d_{\mathrm{eff}}(\alpha_{\min}\mathbf{L}) \rfloor$, we have:

$$d_{\mathrm{eff}}(\alpha_{\max}\mathbf{L}) < d_{\mathrm{eff}}\left((1 + \tfrac{1}{(k+3)^2})\alpha_{\min}\mathbf{L}\right) \leq \left(1 + \tfrac{1}{(k+3)^2}\right) d_{\mathrm{eff}}(\alpha_{\min}\mathbf{L})$$

$$\leq d_{\mathrm{eff}}(\alpha_{\min}\mathbf{L}) + \frac{1}{\lfloor d_{\mathrm{eff}}(\alpha_{\min}\mathbf{L}) \rfloor + 3}.$$

Now, there are two cases. Either $\lfloor d_{\mathrm{eff}}(\alpha_{\max}\mathbf{L}) \rfloor = \lfloor d_{\mathrm{eff}}(\alpha_{\min}\mathbf{L}) \rfloor$, in which case Lemma 11 immediately implies that there are at most two possible modes of the Poisson Binomial $\mathrm{PB}(\alpha\mathbf{L})$ among all values of $\alpha \in [\alpha_{\min}, \alpha_{\max}]$, and they must be achieved by $\alpha_{\min}$ and by $\alpha_{\max}$. If $\lfloor d_{\mathrm{eff}}(\alpha_{\max}\mathbf{L}) \rfloor = \lfloor d_{\mathrm{eff}}(\alpha_{\min}\mathbf{L}) \rfloor + 1$, then the same conclusion is reached by observing that:

$$d_{\mathrm{eff}}(\alpha_{\max}\mathbf{L}) \leq \lfloor d_{\mathrm{eff}}(\alpha_{\max}\mathbf{L}) \rfloor + \frac{1}{\lfloor d_{\mathrm{eff}}(\alpha_{\min}\mathbf{L}) \rfloor + 3} \leq \lfloor d_{\mathrm{eff}}(\alpha_{\max}\mathbf{L}) \rfloor + \frac{1}{\lfloor d_{\mathrm{eff}}(\alpha_{\max}\mathbf{L}) \rfloor + 2},$$

so, by Lemma 11, the mode of $\mathrm{PB}(\alpha_{\max}\mathbf{L})$ must be $\lfloor d_{\mathrm{eff}}(\alpha_{\max}\mathbf{L}) \rfloor$, and once again there are only two possible modes in the interval $\alpha \in [\alpha_{\min}, \alpha_{\max}]$. With high probability, one of these modes must be $k$, which completes the proof. ∎

## C  BLESS-I algorithm

In this section we present the omitted BLESS-I algorithm with proofs of its accuracy and efficiency. For simplicity, in the entirety of this section we will assume that $k \geq 2$. Note that this can be relaxed, at the only cost of slightly more complex constants (e.g., $\alpha_{\mathrm{init}} = \max\{k - 1, 1\}/\mathrm{tr}(\mathbf{L})$) instead of $\alpha_{\mathrm{init}} = (k-1)/\mathrm{tr}(\mathbf{L})$). Moreover, the case $k = 1$ is qualitatively different, as in a 1-DPP the marginal and joint distribution coincide, making it much simpler to sample from.

### C.1  BLESS

We begin by reporting the BLESS algorithm [37] and several of its properties. Note that BLESS was originally introduced as a ridge leverage score (RLS) sampling algorithm. However in the context of DPPs the RLS of an item coincides exactly with its marginal inclusion probability, i.e., $\ell_i(\mathbf{L})$ is the RLS of the $i$-th item. Therefore we can leverage any RLS sampler both to generate dictionaries as well as RLS estimate for $\alpha$-DPP. We choose to use BLESS as a starting point because, to our knowledge, it is the only rescaling-aware RLS sampler existing in the literature. We report BLESS, in its rejection sampling version, in full in Algorithm 5 with the only notational difference of using a rescaling $\alpha \leq 1$ rather than a regularization $\lambda$, with a conversion $\alpha \approx 1/(\lambda n)$ between the two.

**Proposition 4** (Thm. 1 by Rudi et al. [37]). *For some $\alpha' \geq \alpha$, let $\mathcal{D}$ be a dictionary generated using BLESS ran with parameter $q \geq 54\kappa^2 \frac{(2\varepsilon+1)^2}{\varepsilon^2} \log(12n^2/\delta)$. Then w.p. $1 - \delta$ for all $i$,*

- *the dictionary $\mathcal{D}^i$ and weights are $(\varepsilon, \alpha^i)$-accurate,*
- *the approximate marginals $l_j$ computed using $\mathcal{D}^i$ satisfy $\frac{1}{1+\varepsilon}\ell_j(\alpha^i) \leq l_j \leq \frac{1}{1-\varepsilon}\ell_j(\alpha^i)$.*

---
**Algorithm 5** BLESS (rejection-based version)
---
**Input:** $\mathbf{L} \in \mathbb{R}^{n \times n}$, $q > 0$, $k$, $\alpha_{\max}$
 1: Initialize $i = 0$, $\alpha^0 = 1/\text{tr}(\mathbf{L})$, $\widehat{d}_{\text{eff}}(\alpha^0 \mathbf{L}) = \frac{1}{2}(k-1)$
 2: Initialize $\mathcal{D}^0$ by sampling $q\alpha^0 n\kappa^2$ elements $\mathcal{D}^0 \overset{\text{i.i.d.}}{\sim} (1/n, \ldots, 1/n)$ and weight $w_j^0 = 1/(q\alpha^0\kappa^2)$.
 3: **for** $i = \{1, \ldots, \lceil \log_2(\alpha_{\max}/\alpha^0) \rceil\}$ **do**
 4: $\quad$ Set $\alpha^i = 2\alpha^{i-1}$, $b^i = \min\{q\alpha^i\kappa^2, 1\}$
 5: $\quad$ **for** $j = \{1, \ldots, n\}$ **do**
 6: $\quad\quad$ Sample $u_j^i \sim \text{Bernoulli}(b^i)$
 7: $\quad\quad$ **if** $u_j^i = 1$ **then**
 8: $\quad\quad\quad$ Compute $l_j^i$ using Equation 1 and $\mathcal{D}^{i-1}$
 9: $\quad\quad\quad$ Sample $z_j^i \sim \text{Bernoulli}(\min\{ql_j, b^i\}/b^i)$
10: $\quad\quad$ **end if**
11: $\quad$ **end for**
12: $\quad$ Set $\sigma^i = \{j \in [n] : z_j^i = 1\}$, $\mathcal{D}^i = \sigma^i$, $w_j^i = 1/\min\{ql_{\sigma_j^i}, b^i\}$
13: **end for**
14: **return** $\mathcal{D}^{\alpha^{\lceil \log_2(\alpha_{\max}/\alpha^0) \rceil}}$
---

- *the size of the dictionary $m^i = |\mathcal{D}^i|$ is bounded as $d_{\text{eff}}(\alpha^i \mathbf{L})/2 \leq m/q \leq 2d_{\text{eff}}(\alpha^i \mathbf{L})$,*

*and the algorithm runs in* $\mathcal{O}\left((\min\{\alpha_{\max} n\kappa^2, 1\}d_{\text{eff}}(\alpha_{\max}\mathbf{L})^2 \log(n/\delta)^3) \log(\alpha_{\max}\text{tr}(\mathbf{L}))\right)$ *time.*

Note that all results presented in Proposition 4 are only reformulations of Theorem 1 from [37]. The only exception is the lower bound $m/q \geq d_{\text{eff}}(\alpha^i \mathbf{L})/2$, since the original BLESS analysis was only interested in showing that $m/q \leq d_{\text{eff}}(\alpha^i \mathbf{L})/2$. However, the same concentration argument of Lemma 6 in [37] also holds for the lower bound we report here.

## C.2 Modification to BLESS

In order to use BLESS in our approach for DPP sampling, we need to make a few modifications. Compared to BLESS, our BLESS-I (Algorithm 6):

- automatically computes an appropriate $\alpha_{\max}$ rather than taking it as input;
- introduces a novel $\alpha_{\text{init}}$ to initialize $\alpha^0$ that both takes into account the desired DPP size $k$ and is a valid lower bound for the interval search;
- automatically computes an appropriate $\alpha_{\min}$ rather than setting $\alpha_{\min} = \alpha^0$;
- uses the last $d_{\text{eff}}(\alpha_{\max}\mathbf{L})$ estimate to generate a dictionary $\mathcal{D}^{\alpha_{\max}}$ that is guaranteed to be $(1/d_{\text{eff}}(\alpha_{\max}\mathbf{L}), \alpha_{\max})$-accurate.

**Lemma 5** (restated). *For any matrix $\mathbf{L}$ and $0 < \alpha \leq 1$, we have $d_{\text{eff}}(\alpha\mathbf{L})/d_{\text{eff}}(\mathbf{L}) \geq \alpha \geq d_{\text{eff}}(\alpha\mathbf{L})/\text{tr}(\mathbf{L})$.*

*Proof of Lemma 5.* From the definition $d_{\text{eff}}(\alpha\mathbf{L}) = \text{tr}(\alpha\mathbf{L}(\alpha\mathbf{L} + \mathbf{I})^{-1})$. Then the first half comes from

$$\text{tr}(\alpha\mathbf{L}(\alpha\mathbf{L} + \mathbf{I})^{-1}) \leq \text{tr}(\alpha\mathbf{L}(\mathbf{I})^{-1}) = \alpha\text{tr}(\mathbf{L}),$$

while for the second half we have

$$\text{tr}(\alpha\mathbf{L}(\alpha\mathbf{L} + \mathbf{I})^{-1}) \geq \alpha\text{tr}(\mathbf{L}(\mathbf{L} + \mathbf{I})^{-1}) = \alpha d_{\text{eff}}(\mathbf{L}).$$

■

**Lemma 4** (restated). *W.p. $1 - \delta$ BLESS-I runs in time $\widetilde{\mathcal{O}}\left(\min\{\alpha_{\max}\kappa^2, 1\}nk^6 + k^9\right)$ and satisfies:*

1. *The interval $[\alpha_{\min}, \alpha_{\max}]$ is bounded by $\frac{1}{4}(k-1)/\text{tr}(\mathbf{L}) \leq \alpha_{\min} \leq \alpha_{\max} \leq 8(k+2)/d_{\text{eff}}(\mathbf{L})$*
2. *There is $\alpha_\star \in [\alpha_{\min}, \alpha_{\max}]$ for which $k$ is the mode of $|S|$ where $S \sim \text{DPP}(\alpha_\star \mathbf{L})$;*
3. *The dictionary $\mathcal{D}^{\alpha_{\max}}$ satisfies the conditions from Theorem 2 for any $\alpha \in [\alpha_{\min}, \alpha_{\max}]$.*

---

**Algorithm 6** BLESS modified to compute the search interval (BLESS-I)

---

**Input:** $\mathbf{L} \in \mathbb{R}^{n \times n}$, $q > 0$, $k$

1: Initialize $i = 0$, $\alpha^0 = \alpha_{\text{init}} = (k-1)/(n\kappa^2)$, $\widehat{d}_{\text{eff}}(\alpha^0 \mathbf{L}) = \frac{1}{2}(k-1)$

2: Initialize $\mathcal{D}^0$ by sampling $q\alpha^0 n\kappa^2$ elements $\mathcal{D}^0 \overset{\text{i.i.d.}}{\sim} (1/n, \dots, 1/n)$ and weight $w_j^0 = 1/(q\alpha^0\kappa^2)$.

3: **while** $\widehat{d}_{\text{eff}}(\alpha^i \mathbf{L}) \le 2(k+2)$ **do**

4:    Set $i = i+1$, $\alpha^i = 2\alpha^{i-1}$, $\alpha_{\max} = \alpha^i$, $b^i = \min\{q\alpha^i\kappa^2, 1\}$

5:    **for** $j = \{1, \dots, n\}$ **do**

6:      Sample $u_j^i \sim \text{Bernoulli}(b^i)$

7:      **if** $u_j^i = 1$ **then**

8:        Compute $l_j^i$ using Equation 1 and $\mathcal{D}^{i-1}$

9:        Sample $z_j^i \sim \text{Bernoulli}(\min\{ql_j, b^i\}/b^i)$

10:      **end if**

11:    **end for**

12:    Set $\sigma^i = \{j \in [n] : z_j^i = 1\}$, $\mathcal{D}^i = \sigma^i$, $w_j^i = 1/\min\{ql_{\sigma_j^i}, 1\}$

13:    set $\widehat{d}_{\text{eff}}(\alpha^i \mathbf{L}) = |\mathcal{D}^i|/q$

14:    **if** $\widehat{d}_{\text{eff}}(\alpha^{i-1}\mathbf{L}) \le \frac{1}{2}(k-1)$ and $\widehat{d}_{\text{eff}}(\alpha^i \mathbf{L}) > \frac{1}{2}(k-1)$ **then**

15:      Set $\alpha_{\min} = \alpha^{i-1}$

16:    **end if**

17: **end while**

     *Computing approximate RLS.*

18: Set $\mathcal{D}^{\alpha_{\max}} = \emptyset$, $q' = q\widehat{d}_{\text{eff}}(\alpha^i \mathbf{L})^2$, $b^{\max} = \min\{q'\alpha^i\kappa^2, 1\}$

19: **for** $j = \{1, \dots, n\}$ **do**

20:    Sample $u_j^{\max} \sim \text{Bernoulli}(b^{\max})$

21:    **if** $u_j^{\max} = 1$ **then**

22:      Compute $l_j^{\max}$ using Equation 1 and $\mathcal{D}^i$

23:      Sample $z_j^{\max} \sim \text{Bernoulli}(\min\{q'l_j^{\max}, b^{\max}\}/b^{\max})$

24:      If $z_j^{\max} = 1$, add $j$ to $\mathcal{D}^{\alpha_{\max}}$ with weight $w_j^{\alpha_{\max}} = \frac{1}{\min\{q'l_j^{\max}, b^{\max}\}}$

25:    **end if**

26: **end for**

     *Final dictionary construction.*

27: **return** $\alpha_{\min}, \alpha_{\max}, \mathcal{D}^{\alpha_{\max}}$

---

*Proof of Lemma 4.* Throughout the proof we will make use of Proposition 4, in particular that $\frac{1}{2}d_{\text{eff}}(\alpha^i \mathbf{L}) \le \widehat{d}_{\text{eff}}(\alpha^i \mathbf{L}) \le 2d_{\text{eff}}(\alpha^i \mathbf{L})$. Note that by inverting the relationship we also have the reciprocal guarantee $\frac{1}{2}\widehat{d}_{\text{eff}}(\alpha^i \mathbf{L}) \le d_{\text{eff}}(\alpha^i \mathbf{L}) \le 2\widehat{d}_{\text{eff}}(\alpha^i \mathbf{L})$.

**Claim (1): size of the interval.** Applying Lemma 5 we have that $\alpha_{\max} \le d_{\text{eff}}(\alpha_{\max}\mathbf{L})/d_{\text{eff}}(\mathbf{L})$. We need now to further upper bound $d_{\text{eff}}(\alpha_{\max}\mathbf{L})$ BLESS-I's terminating condition (Line 3) only guarantees the lower bound $\widehat{d}_{\text{eff}}(\alpha_{\max}\mathbf{L}) \ge 2(k+2)$. To this end we will use a property of RLS (see Lemma 3 from [37]) that says that if $\alpha^i > \alpha^{i-1}$ then $d_{\text{eff}}(\alpha^i \mathbf{L}) \le \frac{\alpha^i}{\alpha^{i-1}}d_{\text{eff}}(\alpha^{i-1}\mathbf{L})$. In our case, $\alpha^i/\alpha^{i-1} = 2$ and $d_{\text{eff}}(\alpha^i \mathbf{L}) \le 2d_{\text{eff}}(\alpha^{i-1}\mathbf{L})$. Now, let $i$ be the index before the loop exit condition in Algorithm 6 is satisfied (i.e., $\alpha_{\max} = \alpha^{i+1}$). Then we have $\widehat{d}_{\text{eff}}(\alpha^i \mathbf{L}) \le 2(k+2)$, using Proposition 4 we further bound $d_{\text{eff}}(\alpha^i \mathbf{L}) \le 4(k+2)$, which implies that $d_{\text{eff}}(\alpha^{i+1}\mathbf{L}) = d_{\text{eff}}(\alpha_{\max}\mathbf{L}) \le 8(k+2)$. Going back to our bound we obtain $\alpha_{\max} \le d_{\text{eff}}(\alpha_{\max}\mathbf{L})/d_{\text{eff}}(\mathbf{L}) \le 8(k+2)/d_{\text{eff}}(\mathbf{L})$.

The side of $\alpha_{\min}$ is much simpler. From Lemma 5 we have that $\alpha_{\min} \ge d_{\text{eff}}(\alpha_{\min}\mathbf{L})/\text{tr}(\mathbf{L})$, and from the algorithm we know that $\widehat{d}_{\text{eff}}(\alpha_{\min}\mathbf{L}) \ge \frac{1}{2}(k-1)$. Combining this with Proposition 4 we get

$$\alpha_{\min} \ge d_{\text{eff}}(\alpha_{\min}\mathbf{L})/\text{tr}(\mathbf{L}) \ge \tfrac{1}{2}\widehat{d}_{\text{eff}}(\alpha_{\min}\mathbf{L})/\text{tr}(\mathbf{L}) \ge \tfrac{1}{4}(k-1)/\text{tr}(\mathbf{L}).$$

**Claim (2): validity of the interval.** To begin, remember from Lemma 11 that the mode $m_\alpha$ of the sample size of $\text{DPP}(\alpha\mathbf{L})$ is bounded by $\lfloor d_{\text{eff}}(\alpha\mathbf{L})\rfloor \le m_\alpha \le \lfloor d_{\text{eff}}(\alpha\mathbf{L})\rfloor + 1$. To guarantee the validity of our interval, we show that $m_{\alpha_{\min}} \le k$, and $m_{\alpha_{\max}} \ge k+1$. Due to the monotonicity of the mode of a Poisson Binomial distribution (see Lemma 10) we have therefore that starting from $m_{\alpha_{\min}}$ the mode increases with $\alpha$, until it reaches $k$ for some $\alpha_\star \in [\alpha_{\min}, \alpha_{\max}]$, and then continue increasing until it reaches $k+1 \le m_{\alpha_{\max}}$.

Figure 4: Fraction of items observed by $\alpha$-DPP on the small scale experiment.

Figure 5: Large scale experiment using linear similarity.

Concretely, we have that

$$m_{\alpha_{\min}} \overset{\text{Lemma 11}}{\leq} d_{\text{eff}}(\alpha_{\min}\mathbf{L}) + 1 \overset{\text{Proposition 4}}{\leq} 2 \cdot \widehat{d}_{\text{eff}}(\alpha_{\min}\mathbf{L}) + 1 < 2 \cdot \tfrac{1}{2}(k-1) + 1 = k,$$

where the last inequality is due to the condition from Line 14 in BLESS-I. Similarly

$$m_{\alpha_{\max}} \overset{\text{Lemma 11}}{\geq} d_{\text{eff}}(\alpha_{\max}\mathbf{L}) - 1 \overset{\text{Proposition 4}}{\geq} \tfrac{1}{2} \cdot \widehat{d}_{\text{eff}}(\alpha_{\max}\mathbf{L}) - 1 > \tfrac{1}{2} \cdot 2(k+2) - 1 = k+1,$$

where this time the last inequality is due to the condition from Line 3 in BLESS-I. Finally, we have to guarantee that $\alpha_{\text{init}}$ is also a valid lower bound, or we will never able to correctly set $\alpha_{\min}$. This is easy to show using Lemma 5

$$m_{\alpha_{\text{init}}} \leq d_{\text{eff}}(\alpha_{\text{init}}\mathbf{L}) + 1 \overset{\text{Lemma 5}}{\leq} \alpha_{\text{init}}\text{tr}(\mathbf{L}) + 1 = \tfrac{\text{tr}(\mathbf{L})}{n\kappa^2}(k-1) + 1 \leq k - 1 + 1 = k,$$

making it a valid initialization for the lower bound.

**Claim (3): quality of $\mathcal{D}^{\max}$.** At the end of the main loop, due to Proposition 4 we have that $\widehat{d}_{\text{eff}}(\alpha^i\mathbf{L}) \geq \tfrac{1}{2}d_{\text{eff}}(\alpha^i\mathbf{L})$, and that since $\alpha^i = 2\alpha^{i-1} = 2\alpha_{\max}$, $d_{\text{eff}}(\alpha^i\mathbf{L}) \geq d_{\text{eff}}(\alpha_{\max}\mathbf{L})$. Therefore, setting $q' = 4\widehat{d}_{\text{eff}}(\alpha^i\mathbf{L})^2 q$ is sufficient to invoke Proposition 4 with $\varepsilon = 1/d_{\text{eff}}(\alpha_{\max}\mathbf{L})$ and obtain an $(1/d_{\text{eff}}(\alpha_{\max}), \alpha_{\max})$-accurate dictionary. Moreover, it is easy to see that for any $\alpha' \geq \alpha$ and $\varepsilon' \leq \varepsilon$, an $(\varepsilon', \alpha')$-accurate dictionary is also an $(\varepsilon, \alpha)$-accurate dictionary (see Proposition 5). Since $\alpha_{\max} \geq \alpha$ for the whole duration of the binary search, and therefore $d_{\text{eff}}(\alpha_{\max}\mathbf{L}) \geq d_{\text{eff}}(\alpha\mathbf{L})$, our $\mathcal{D}^{\max}$ dictionary is sufficiently accurate for the whole duration of the binary search. ∎

# D  Additional experimental details

Both DPP-VFX and $\alpha$-DPP rely on BLESS or BLESS-I to generate their input dictionaries. For this preprocessing phase, the major hyperparameters to tune are $q_{\text{BLESS}}$ and $q_{\text{dpp}}$, i.e., the $q$ and $q'$ parameters indicated in Algorithm 6.[7]

Note that theory suggests to set $q_{\text{BLESS}} \approx \mathcal{O}(\log(n))$ and $q_{\text{dpp}} \approx \mathcal{O}(d_{\text{eff}}(\alpha\mathbf{L})^2)$, but they can be freely tuned since both $\alpha$-DPP and DPP-VFX remain exact samplers for any hyperparameter choice. However, $q_{\text{BLESS}}$ and $q_{\text{dpp}}$ do impact acceptance rate and runtime, and even more importantly too low values can result in empty dictionaries which force the algorithm to be stopped.

In our case, we start with $q_{\text{BLESS}} = 2$ and $q_{\text{dpp}} = 2$, and increase them until the DPPy implementation does not return an empty dictionary. We also keep the same value for $\alpha$-DPP and DPP-VFX so that for similar $\alpha$ they operate with similarly accurate and large dictionaries. The final values are $q_{\text{BLESS}} = 5$ and $q_{\text{dpp}} = 10$ for the small scale experiment (Figure 1), and $q_{\text{BLESS}} = 4$ and $q_{\text{dpp}} = 5$ for the large scale experiment (Figure 2).

For completeness, in addition to the fraction of observed items in the large scale experiment (Figure 3), we also report the fraction of observed items in the small scale experiment (Figure 4). We note that, for the small scale experiment, until $n$ exceeds 10000, $\alpha$-DPP is still observing all items, and only when the item collection becomes sufficiently large uniform sampling starts to play a role.

Finally, we report another experiment taken directly from the benchmark of Dereziński et al. [13] where a linear similarity is used instead of `rbf` similarity. We see that in this setting $d_{\mathrm{eff}}(\mathbf{L})$ grows slower with $n$, since the similarity/kernel is less expressive. As a consequence the gap between $\alpha$-DPP and DPP-VFX (i.e., the advantage of using uniform intermediate sampling) is reduced, but remains impactful.

## E  Miscellaneous proofs

In this section we present omitted miscellaneous facts and proofs for completeness.

**Definition 2.** *Given a psd matrix $\mathbf{L}$, its $i$th ridge leverage score (RLS) $\ell_i(\mathbf{L})$ is the $i$th diagonal entry of $\mathbf{L}(\mathbf{I} + \mathbf{L})^{-1}$. The sum $\sum_{i=1}^{n} \ell_i(\mathbf{L}) = d_{\mathrm{eff}}(\mathbf{L})$ of the RLSs is equal to the effective dimension of $\mathbf{L}$.*

**Definition 3** ([2, 5]). *A dictionary $\mathcal{D}$ and its associated weighting matrix $\mathbf{W}$ are $(\varepsilon, \alpha)$-accurate if $\|\alpha\mathbf{L}(\mathbf{I} + \alpha\mathbf{L})^{-1}(\mathbf{I} - \overline{\mathbf{W}})\| \le \varepsilon$, where $\overline{\mathbf{W}} \in \mathbb{R}^{n \times n}$ is diagonal with $\overline{\mathbf{W}}_{i,i} = \sum_{j=1}^{m} w_j \mathbb{I}\{\mathcal{D}_j = i\}$.*

**Proposition 5** ([2, 5]). *A dictionary $\mathcal{D}$ and its associated weighting matrix $\mathbf{W}$ are $(\varepsilon, \alpha)$-accurate if*

$$\|(\mathbf{I} + \alpha\varphi([n])^\top\varphi([n]))^{-1/2}(\alpha\varphi([n])^\top\varphi([n]) - \alpha\varphi(\mathcal{D})^\top\mathbf{W}\varphi(\mathcal{D}))(\mathbf{I} + \alpha\varphi([n])^\top\varphi([n]))^{-1/2}\| \le \varepsilon,$$

*or equivalently*

$$\|(\mathbf{I}/\alpha + \varphi([n])^\top\varphi([n]))^{-1/2}(\varphi([n])^\top\varphi([n]) - \varphi(\mathcal{D})^\top\mathbf{W}\varphi(\mathcal{D}))(\mathbf{I}/\alpha + \varphi([n])^\top\varphi([n]))^{-1/2}\| \le \varepsilon,$$

*or yet equivalently*

$$(1 - \varepsilon)(\mathbf{I}/\alpha + \varphi([n])^\top\varphi([n])) \preceq \mathbf{I}/\alpha + \varphi(\mathcal{D})^\top\mathbf{W}\varphi(\mathcal{D}) \preceq (1 + \varepsilon)(\mathbf{I}/\alpha + \varphi([n])^\top\varphi([n])).$$

Note that using Proposition 5 it is easy to see that for any $\alpha' \ge \alpha$ and $\varepsilon' \le \varepsilon$, an $(\varepsilon', \alpha')$-accurate dictionary is also an $(\varepsilon, \alpha)$-accurate dictionary since $\mathbf{I}/\alpha' \preceq \mathbf{I}/\alpha$ and therefore

$$\|(\mathbf{I}/\alpha + \varphi([n])^\top\varphi([n]))^{-1/2}(\varphi([n])^\top\varphi([n]) - \varphi(\mathcal{D})^\top\mathbf{W}\varphi(\mathcal{D}))(\mathbf{I}/\alpha + \varphi([n])^\top\varphi([n]))^{-1/2}\|$$
$$\le \|(\mathbf{I}/\alpha' + \varphi([n])^\top\varphi([n]))^{-1/2}(\varphi([n])^\top\varphi([n]) - \varphi(\mathcal{D})^\top\mathbf{W}\varphi(\mathcal{D}))(\mathbf{I}/\alpha' + \varphi([n])^\top\varphi([n]))^{-1/2}\|$$
$$\le \varepsilon' \le \varepsilon.$$

Moreover, using basic algebraic manipulation we can see that for any matrix/operator $\mathbf{A}$ we have

$$(\mathbf{I} + \mathbf{A}\mathbf{A}^\top)^{-1} = \mathbf{I} - \mathbf{A}(\mathbf{I} + \mathbf{A}^\top\mathbf{A})^{-1}\mathbf{A}^\top,$$

which applied to $\mathbf{A} = \sqrt{\alpha}\mathbf{W}^{1/2}\varphi(\mathcal{D})$ gives us the following reformulation from [5, 37]:

$$\begin{aligned}
l_i &= \alpha[\mathbf{L} - \mathbf{L}_{[n],\mathcal{D}}^\top(\alpha\mathbf{L}_{\mathcal{D},\mathcal{D}} + \mathbf{W}^{-1})^{-1}\mathbf{L}_{[n],\mathcal{D}}]_{i,i} \\
&= \alpha[\varphi([n])\varphi([n])^\top - \alpha\varphi([n])\varphi(\mathcal{D})^\top(\alpha\varphi(\mathcal{D})\varphi(\mathcal{D})^\top + \mathbf{W}^{-1})^{-1}\varphi(\mathcal{D})\varphi([n])^\top]_{i,i} \\
&= \alpha[\varphi([n])\left(\mathbf{I} - \alpha\varphi(\mathcal{D})^\top(\alpha\varphi(\mathcal{D})\varphi(\mathcal{D})^\top + \mathbf{W}^{-1})^{-1}\varphi(\mathcal{D})\right)\varphi([n])^\top]_{i,i} \\
&= \alpha[\varphi([n])\left(\mathbf{I} - \alpha\varphi(\mathcal{D})^\top\mathbf{W}^{1/2}(\alpha\varphi(\mathcal{D})\mathbf{W}\varphi(\mathcal{D})^\top + \mathbf{I})^{-1}\mathbf{W}^{1/2}\varphi(\mathcal{D})\right)\varphi([n])^\top]_{i,i} \\
&= \alpha[\varphi([n])(\mathbf{I} + \alpha\varphi(\mathcal{D})^\top\mathbf{W}\varphi(\mathcal{D}))^{-1}\varphi([n])^\top]_{i,i} \\
&= \alpha\varphi(i)^\top(\mathbf{I} + \alpha\varphi(\mathcal{D})^\top\mathbf{W}\varphi(\mathcal{D}))^{-1}\varphi(i).
\end{aligned}$$

Applying Proposition 5 to the reformulation it is easy to see that

$$\alpha\varphi(i)^\top(\mathbf{I} + \alpha\varphi(\mathcal{D})^\top\mathbf{W}\varphi(\mathcal{D}))^{-1}\varphi(i) = \varphi(i)^\top(\mathbf{I}/\alpha + \varphi(\mathcal{D})^\top\mathbf{W}\varphi(\mathcal{D}))^{-1}\varphi(i)$$
$$\le \tfrac{1}{1-\varepsilon}\varphi(i)^\top(\mathbf{I}/\alpha + \varphi([n])^\top\varphi([n]))^{-1}\varphi(i) = \tfrac{1}{1-\varepsilon}\alpha\varphi(i)^\top(\mathbf{I} + \alpha\varphi([n])^\top\varphi([n]))^{-1}\varphi(i) = \ell_i(\mathbf{L}).$$

**Caching strategy.** Note that if we invoke $\alpha$-DPP multiple times for a fixed $\alpha$, we do not need to recompute all approximations $l_i$ from scratch each time. Rather, we first store an eigendecomposition

of $\widehat{\mathbf{L}}$ to be able to quickly compute $(\alpha\mathbf{L}_{\mathcal{D},\mathcal{D}} + \mathbf{W}^{-1})^{-1}$ in quadratic rather than cubic time. Then, for each item $i$ we store a *cache* of the current upper bound, which is initialized to $\alpha\kappa^2$ and then lowered to $l_i$ when $l_i$ is actually computed. This way we never need to recompute the same $l_i$ twice, and the runtime improves. In particular, computing a single marginal $l_i$ requires $\mathcal{O}(k^6)$ time. So, if all $l_i$ were computed from scratch, then the inner loop of Algorithm 1 would require $\alpha_{\max}\kappa^2 kn \cdot k^6$ to compute $\alpha_{\max}\kappa^2 kn$ marginals $l_i$, one for each item in $\rho$. On the other hand, computing all $l_i$ for all items once and for all would require $n \cdot k^6$ time, and then sampling would be near-constant time using an appropriate multinomial sampler (see [13]). In our case, using the caching strategy we can get the best of both worlds $\widetilde{\mathcal{O}}(\min\{\alpha_{\max}\kappa^2 k, 1\} \cdot n \cdot k^6)$ since we never compute any $l_i$ more than once.