[Reviews · NeurIPS 2020]

Review 1

Summary and Contributions: This paper proposes an algorithm for sampling from k-DPP and demonstrates its effectiveness both with theoretical guarantee and empirical experiments. The proposed algorithm first samples a subset with uniform distribution, compute marginal probabilities in the subset, and then sample a subset of size k. This is in sharp contrast to the state-of-the-art, which relies on the marginal probabilities for all items. The computational complexity is improved from O(n k^10+k^15) to O((n k^6 + k^9) k^0.5), where n is the size of the ground set. Numerical experiments also confirms the superiority of the proposed algorithm.

Strengths: The proposed algorithm has clear and significant benefits over the state-of-the-art, and sampling from k-DPP is important and relevant in machine learning applications. Also, the ideas used in the proposed algorithm can be useful in developing further improvement in k-DPP sampling.

Weaknesses: I do not find much weakness in the paper, but the successful applications of DPPs in machine learning is still limited, although they are growing, and it is unclear at this moment how much impact this paper can have. Also, for a moderate size of the ground set (<100,000), the proposed algorithm performs not much different from the-state-of-the-art.

Correctness: Although I have not checked the proofs, numerical results sufficiently support the correctness of the results.

Clarity: The paper is well written, and the main ideas are clearly delivered.

Relation to Prior Work: Very clear.

Reproducibility: Yes

Additional Feedback: Table 1 does not discuss that the bound for the proposed algorithm holds with probability 1 - delta, while I assume that n^3 bound is with probability 1. In Figure 1, the proposed algorithm and DPP-VFX take ~50 seconds even with very small n. It would help to discuss why these methods takes nontrivial time with very small n. -- Thank you very much for the clarifications in the rebuttal. I look forward to seeing an improved version in the conference.


Review 2

Summary and Contributions: This paper proposes a method which can efficiently select a subset of size "k" to represent a large size of data.

Strengths: 1. Well organized. 2. Detailed proofs.

Weaknesses: The effectiveness of the proposed method needs to be further evaluated.

Correctness: Yes.

Clarity: Generally well written.

Relation to Prior Work: Yes.

Reproducibility: Yes

Additional Feedback: This paper proposes a method which can efficiently select a subset of size "k" to represent a large size of data. This paper is generally well written and the proofs are detailed. From the experiments we can observe the efficiency of the proposed method. My major concern is: As mentioned in the abstract, previous naive heuristic method can not guarantee if the selected items are representative. How can the proposed method ensure that the selected samples resemble the target distribution? I think this should be further evaluated in the experiment section. For example, plot figures using algorithm like t-SNE or compute the distributions using some statistical metrics.


Review 3

Summary and Contributions: This paper presents a scalable sampling algorithm for k-DPPs (determinantal point processes). While a standard DPP places probability mass on subsets up to size n (where n is the ground set size), a k-DPP places mass only on subsets of size k. The main contribution is a new algorithm for DPP sampling, called an alpha-DPP sampler, which samples from a DPP rescaled by a parameter alpha <= 1. This algorithm leverages uniform intermediate sampling to compute only a small subset of the marginals for the items in the ground set, which is much faster than recent prior work that requires computing the full collection of item marginals (or other, even more expensive alternatives). Experiments indicated that the proposed algorithm is substantially faster than a state-of-the-art k-DPP sampler for k-DPPs with very large ground sets.

Strengths: - This is a strong paper. While it leverages aspects of the DPP-VFX sampling algorithm, the proposed k-DPP sampling algorithm appears to be novel, scalable, and well grounded from a theoretical standpoint. - The proposed algorithm is substantially (up to several orders of magnitude) faster than recent prior work on k-DPP sampling (DPP-VFX), which is a significant contribution that can open up k-DPP sampling to applications that involve much larger ground sets.

Weaknesses: - While each component of the proposed sampling algorithm is described clearly (including Alg. 5, Alg. 2, and Alg. 1), the full k-DPP sampling algorithm that uses all of these components does not appear to be concisely presented in a single location in this paper. Even though these components appear to be used together in a relatively straightforward fashion, the full sampling should be clearly and concisely presented, preferably in an additional algorithmic or pseudocode block. - Fig. 5 in the supplementary material shows the experimental results comparing alpha-DPP with DPP-VFX for a DPP kernel with linear similarity, rather than rbf similarity used for the experiment shown in Fig. 2 in the main paper. Comparing the two, the runtime improvement for alpha-DPP over DPP-VFX is substantially smaller for the linear similarity case, indicating that the alpha-DPP sampling algorithm is sensitive to the distribution of the item marginals and the properties of the kernel. This is a potential limitation of this work.

Correctness: The claims and empirical methodology appear to be correct.

Clarity: The paper is well written, although the presentation of the full k-DPP sampling algorithm should be improved (see “weaknesses” above for details).

Relation to Prior Work: A clear discussion of related work on k-DPP and DPP sampling, including issues with existing DPP sampling algorithms such as DPP-VFX, and how this work differs from prior work, is provided in Sections 1 and 2.

Reproducibility: Yes

Additional Feedback: How does alpha-DPP perform, compared to DPP-VFX, when sampling from a DPP kernel learned from real data? It would be interesting to include the results of such an experiment in a future version of this paper, since k-DPP sampling may be used in settings where a DPP kernel is learned. Would it be possible to extend the alpha-DPP sampling approach presented in this paper to sampling from a standard DPP, rather than a k-DPP? That is, could this approach form the basis for a method for sampling from a standard DPP without looking at all items? -- Post-rebuttal update: I've read the other reviews and the authors' rebuttal. I'm satisfied that the rebuttal has addressed the questions and comments from the reviewers, and hope to see a revised version of this paper at the conference. I maintain my overall score of an 8.

[Author Response · NeurIPS 2020]

We thank all the reviewers for their comments and feedback.

First, we would like to address Reviewer #3's concerns about the quality of the samples generated by our algorithm.
In this paper, we propose an efficient algorithm for sampling from an established family of distributions that select
representative samples of data: determinantal point processes of size $k$, a.k.a., $k$-DPPs. The effectiveness of $k$-DPPs at
producing representative samples has been shown by extensive prior work at top ML conferences (see the first paragraph
of Section 1). In our paper, we formally prove that our algorithm samples *exactly from the target k-DPP distribution*.
Thus, the quality of the samples produced by the algorithm is established by all the prior work on $k$-DPPs.

**To Reviewer #2**

*"Table 1 does not discuss that the bound for the proposed algorithm holds with probability* $1 - \delta$*, while I assume that*
$n^3$ *bound is with probability 1."*
Thanks for pointing this out. We will clarify that the eigendecomposition-based methods, although much slower, have a
deterministically bounded runtime.

*"In Figure 1, the proposed algorithm and DPP-VFX take ~50 seconds even with very small* $n$*. It would help to discuss*
*why these methods takes nontrivial time with very small* $n$*."*

The higher complexity at low $n$ is due to the search and rejection
steps in the $k$-DPP sampler. Specifically, $\alpha$-DPP samples from a
$k$-DPP by repeatedly sampling from a random-size DPP until it
generates a sample with size exactly $k$. Our theory ensures that
this will happen after only a small number of rejections. In the
experiments from Figure 1 in the paper, this amounts to roughly
8 to 10 rejections until acceptance (see Figure 1 in this response).
A single DPP sample requires $\sim 5$ seconds to generate using
$\alpha$-DPP, resulting in a runtime around 50 seconds for the $k$-DPP.
Note however that this is mostly because we do not choose a
different oversampling parameter for different sample sizes. In
particular, for small $n$ we can increase the oversampling both in
the Nyström approximation as well as in the intermediate sample
size to reduce the number of rejections and improve overall
runtime. We will include this discussion in the final version.

Figure 1: Number of rejections when converting a
random-size DPP into a $k$-DPP.

**To Reviewer #3**

*"As mentioned in the abstract, previous naive heuristic method can not guarantee if the selected items are representative.*
*How can the proposed method ensure that the selected samples resemble the target distribution?"*
Theorem 1 clearly states that the samples returned by $\alpha$-DPP exactly follow the target (i.e. $k$-DPP) distribution. That is,
we rigorously prove (Lemma 6 in Appendix A) that $\alpha$-DPP's induced distribution does not simply resemble a $k$-DPP, or
is $\varepsilon$-close like an MCMC sampler, but strictly follows the definition of a $k$-DPP distribution.

*"I think this should be further evaluated in the experiment section. For example, plot figures using algorithm like t-SNE*
*or compute the distributions using some statistical metrics."*
We *formally prove* that the two distributions are equal. Any statistical discrepancy we could find by running an
experiment would only be a statistical error due to the finite sample size.

**To Reviewer #4**

*"...the full sampling should be clearly and concisely presented, preferably in an additional algorithmic or pseudocode*
*block."*
We will move the paragraph on caching strategy in Appendix E earlier in the appendix, and add more details on the full
algorithmic pipeline. Note that in addition we provide in the supplementary material a reference implementation in
python that also shows how to integrate the whole algorithm.

*"Would it be possible to extend the* $\alpha$*-DPP sampling approach presented in this paper to sampling from a standard DPP,*
*rather than a* $k$*-DPP?"*
This can be done, and it is indeed already the case, in that, $\alpha$-DPP (Algorithm 1) by itself samples from a standard DPP
but rescaled by a constant factor, which is then used to construct $k$-DPP samples. More generally, $\alpha$-DPP can sample
from a standard DPP without looking at all elements whenever it has a prior upper bound on the marginal inclusion
probabilities (see "Beyond uniform sampling" on L222). Rescaling provides such a bound for $k$-DPPs. We hope that, as
the usage of $\alpha$-DPP increases, more structured bounds will be discovered and integrated in the algorithm.

[Meta-Review · NeurIPS 2020]

The reviewers found this paper interesting and well written. This is a nice contribution to NeurIPS.